# Microglial trogocytosis and the complement system regulate axonal pruning in vivo

**Tony KY Lim, Edward S Ruthazer***

Department of Neurology & Neurosurgery, Montreal Neurological Institute-Hospital, McGill University, Montreal, Canada

**Abstract** Partial phagocytosis—called trogocytosis—of axons by microglia has been documented in ex vivo preparations but has not been directly observed in vivo. The mechanisms that modulate microglial trogocytosis of axons and its function in neural circuit development remain poorly understood. Here, we directly observe axon trogocytosis by microglia in vivo in the developing *Xenopus laevis* retinotectal circuit. We show that microglia regulate pruning of retinal ganglion cell axons and are important for proper behavioral response to dark and bright looming stimuli. Using bioinformatics, we identify amphibian regulator of complement activation 3, a homolog of human CD46, as a neuronally expressed synapse-associated complement inhibitory molecule that inhibits trogocytosis and axonal pruning. Using a membrane-bound complement C3 fusion protein, we demonstrate that enhancing complement activity enhances axonal pruning. Our results support the model that microglia remodel axons via trogocytosis and that neurons can control this process through expression of complement inhibitory proteins.

**\*For correspondence:**
edward.ruthazer@mcgill.ca

**Competing interests:** The authors declare that no competing interests exist.

## Introduction

Microglia, the immune cells of the CNS, are vital for the maintenance and development of a healthy brain. Constantly surveilling the brain (*Nimmerjahn et al., 2005*; *Wake et al., 2009*), these highly phagocytic cells are thought to contribute to developmental synaptic remodeling by phagocytosing inappropriate or supernumerary synapses, a hypothesis that has derived considerable support from histological and immunohistochemical evidence identifying synaptic components within endocytic compartments in microglia (*Paolicelli et al., 2011*; *Schafer et al., 2012*; *Stevens et al., 2007*; *Tremblay et al., 2010*). This hypothesis is further supported by numerous studies demonstrating that microglia depletion leads to exuberant axonal outgrowth (*Pont-Lezica et al., 2014*; *Squarzoni et al., 2014*), impaired pruning of excess synapses (*Ji et al., 2013*; *Milinkeviciute et al., 2019*), and increased spine density (*Wallace et al., 2020*) during development.

Studies depleting microglia or disrupting microglial function provide indirect evidence to support the hypothesis that microglia remodel synapses through phagocytic mechanisms. An inherent weakness of indirect approaches is that the source of the synaptic material within microglia is unknown. For example, it is possible that synaptic components may be found within microglia due to clearance of apoptotic neurons rather than synaptic pruning. More direct approaches are required to verify whether microglia collect synaptic material from living neurons. Currently, direct evidence of complete elimination of synapses by microglial engulfment remains elusive. However, instead of removing entire synapses, microglia have been documented engaging in trogocytosis, or partial elimination, of axons and presynaptic boutons in ex vivo organotypic cultures and in fixed brain tissue using electron microscopy (*Weinhard et al., 2018*), although it remains to be seen whether microglial trogocytosis of axons is a phenomenon that occurs in vivo.

Even if we accept the hypothesis that microglia trogocytose the axonal compartment, many questions remain. What impact does partial elimination of presynaptic structures have on circuit remodeling? It is unclear if this phenomenon is required for proper wiring of neuronal circuits. Does microglial trogocytosis of axons affect axon morphology? While disrupting microglial function enhances axon tract outgrowth (*Pont-Lezica et al., 2014*; *Squarzoni et al., 2014*), it is unknown if this result is because of a disruption in microglial trogocytosis, or whether non-phagocytic mechanisms are in play. Is axonal trogocytosis by microglia mechanistically similar to complement-mediated synaptic pruning? There is extensive evidence demonstrating that the complement system regulates synaptic pruning by microglia via the complement protein C3 (*Paolicelli et al., 2011*; *Schafer et al., 2012*; *Stevens et al., 2007*). However, knockout (KO) mice lacking complement receptor type 3 (CR3), the receptor for activated C3, do not exhibit a deficit in microglial trogocytosis (*Weinhard et al., 2018*), raising the possibility that microglial trogocytosis of axons is mechanistically distinct from complement-mediated synaptic pruning.

In this study, we addressed these questions and directly observed in vivo trogocytosis of retinal ganglion cell (RGC) axons by individual microglial cells in real-time using the developing *Xenopus laevis* retinotectal circuit. We then developed an assay to monitor microglial trogocytosis of axons among the population of microglia in the optic tectum. To investigate the functional role of microglial trogocytosis, we depleted microglial cells and found that this enhanced axon arborization and reversed the behavioral responses to dark and bright looming stimuli. We identified amphibian regulator of complement activation 3 (aRCA3) (*Oshiumi et al., 2009*), a homolog of mammalian CD46, as an endogenously expressed, synapse-associated, complement inhibitory molecule in *Xenopus laevis* RGC neurons. Overexpression of aRCA3 inhibited trogocytosis and axonal pruning. Conversely, expression of a membrane-bound complement C3 fusion protein in RGCs enhanced axonal pruning. Our findings provide direct in vivo evidence supporting the hypothesis that microglia trogocytose presynaptic axonal elements and supports a model in which microglial trogocytosis regulates axonal pruning to promote proper neural wiring during development (*Schafer et al., 2012*; *Weinhard et al., 2018*). In this model, neurons exert local control of microglial trogocytosis and axonal pruning by expressing complement regulatory proteins.

## Results

### Microglia in *Xenopus laevis* tadpoles resemble neonatal mammalian microglia

Microglia in albino *Xenopus laevis* tadpoles were labeled with IB4-isolectin conjugated fluorophores for in vivo imaging. IB4-isolectin binds the RET receptor tyrosine kinase on microglial cells (*Boscia et al., 2013*), but does not lead to production of tumor necrosis factor-α or alterations in microglial morphology (*Grinberg et al., 2011*). Injection of fluorophore conjugated IB4-isolectin into the third ventricle of *Xenopus laevis* tadpoles labels highly mobile cells (*Figure 1A* and *Video 1A*) that have both ameboid-like and primitive ramified-like morphologies, resembling the morphology of embryonic microglial cells. Ameboid microglia are round or irregular cells with filopodia and/or pseudopodia, while primitive ramified microglia have scantly developed, poorly branching processes (*Dalmau et al., 1997*; *Dalmau et al., 1998*). Interestingly, IB4-isolectin-labeled cells can be observed switching back and forth between ameboid-like and primitive ramified-like morphologies (*Figure 1B* and *Video 1B*), suggesting that microglial morphology during development may be more dynamic than previously thought. Microglial cell bodies in the adult brain demonstrate low mobility (migration), while microglia in developing brain have highly mobile cell bodies (*Smolders et al., 2019*). Microglial mobility was determined to be 1.7 ± 0.3 µm/min (mean ± SD) (*Figure 1C*), comparable to what has been observed in neonatal rodent slice culture (1.4 µm/min at E15.5) (*Smolders et al., 2017*). To provide functional confirmation that IB4-isolectin-labeled cells are microglial cells, a laser irradiation injury was performed on the neuropil region. As expected of microglial cells, IB4-isolectin-labeled cells responded to injury by surrounding the damaged zone and removing injured tissue (*Figure 1D* and *Video 2*).

In developing zebrafish larvae, microglia primarily localize to the cell body layer of the optic tectum and are largely excluded from the tectal neuropil (*Svahn et al., 2013*). Conversely, microglia in developing mammalian models are found in neuropil regions (*Dalmau et al., 1997*; *Hoshiko et al.,*

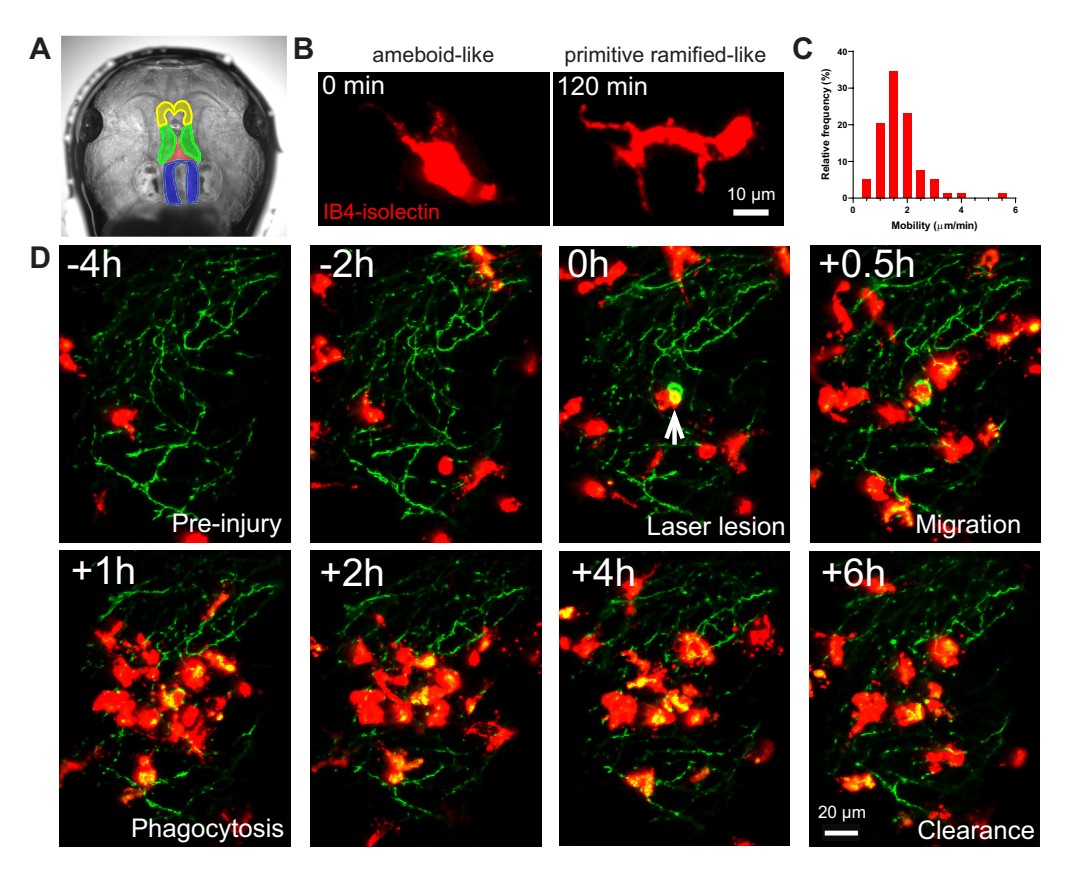

**Figure 1.** IB4-isolectin-conjugated fluorophores label microglial cells in developing *Xenopus laevis* tadpoles. (**A**) The tadpole brain colorized for identification (yellow = olfactory bulb and forebrain, green = optic tectum, blue = hindbrain). To label microglia, IB4-isolectin conjugated fluorophores are injected into the 3rd ventricle (red). (**B**) Dynamic cells with both ameboid-like and primitive ramified-like morphologies are labeled by IB4-isolectin. (**C**) The distribution in mobility of IB4-isolectin-labeled cells under normal conditions. Average velocity is 1.8 ± 0.5 µm/min (mean ± SD, n = 5). (**D**) Laser irradiation injury of the neuropil and response by IB4-isolectin-labeled cells. Laser irradiation injury induced a region of damaged, autofluorescent, tissue. IB4-isolectin-labeled cells mobilize to the injury site and remove the injured tissue by phagocytosis. IB4-isolectin-labeled cells are shown in red, and eGFP-labeled RGC axons are shown in green.

The online version of this article includes the following source data for figure 1:

**Source data 1.** Microglia mobility measurements in *Figure 1C*.

*2012*; *Tremblay et al., 2010*). In the *Xenopus laevis* retinotectal circuit, RGC axons project to the contralateral optic tectum, where they arborize and synapse on tectal neurons in the neuropil region (*Figure 2A*). To examine whether Xenopus microglia interact with the tectal neuropil, RGC axons innervating the neuropil were labeled by bulk electroporation with a plasmid encoding pH-stable green fluorescent protein (pHtdGFP). In vivo live imaging revealed that, microglia in developing *Xenopus laevis* can be found in both the cell body layer and the neuropil region (*Video 3A*). Microglia move in and out of the neuropil region from the cell body layer (*Figure 2B* and *Video 3B*) and move freely through the neuropil (*Figure 2C* and *Video 3B*). Additionally, microglia extend processes into the neuropil to contact axons, with interactions ranging from minutes to hours in duration (*Figure 2D* and *Video 3C*).

## In vivo imaging of RGC neurons reveals microglial trogocytosis of axons and presynaptic structures

We then sought to examine whether microglia cells engage in trogocytosis of RGC axons in the developing *Xenopus laevis* retinotectal circuit. As endosomal organelles are typically acidified (*Casey et al., 2010*), pH-stability of dyes and fluorescent proteins is an important consideration

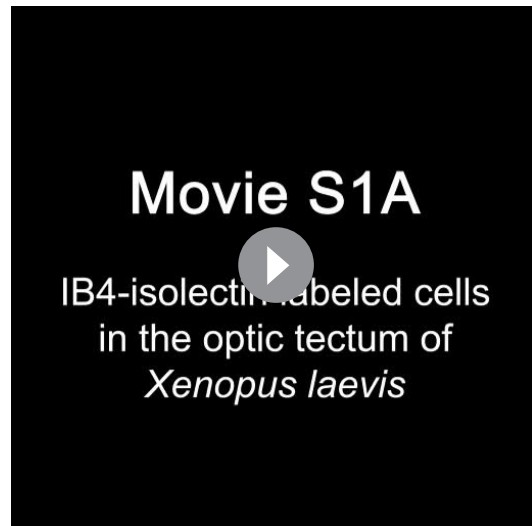

**Video 1.** IB4-isolectin-labeled cells in *Xenopus laevis* tadpoles are morphologically dynamic and highly mobile. (A) IB4-isolectin cells are highly mobile. Timestamp = HH:MM. (B) IB4-isolectin labeled cells have a dynamic morphology, switching back and forth between ameboid-like and primitive ramified-like morphologies.
https://elifesciences.org/articles/62167#video1

when performing live imaging of trogocytosis (*Shinoda et al., 2018*). To reduce quenching of fluorescence, we used pHtdGFP (pKa = 4.8) (*Roberts et al., 2016*) which is more pH-stable than EGFP (pKa = 6.15). We expressed pHtdGFP in RGC axons by electroporation in the eye and labeled microglia with Alexa 594-conjugated IB4-isolectin. Two-photon live imaging revealed that the amount of green fluorescence associated with individual microglial cells increased following interactions with pHtdGFP-labeled axons. In the example shown in *Video 4A, a* microglial cell increases in green fluorescence by twofold following interaction with a pHtdGFP-labeled axon (*Figure 2—figure supplement 1A*). Similarly, in the example shown in *Figure 2E* and *Video 4B*, the green fluorescence in the microglial cell increased threefold following interaction with a pHtdGFP-labeled axon (*Figure 2—figure supplement 1B*). The real-time increase in microglial green fluorescence suggests direct transfer of fluorescent protein from the pHtdGFP-labeled axon and provides direct in vivo evidence of microglial trogocytosis of the presynaptic RGC axon.

Because microglia in developing *Xenopus laevis* tadpoles are highly mobile, the possibility that they may leave the imaging field complicates measuring trogocytosis of axons by individual microglia. To quantify microglial trogocytosis, we instead took the approach of measuring microglial green fluorescence across the population of microglia within the optic tectum. A greater number of pHtdGFP-labeled axons in the optic tectum is expected to lead to more frequent opportunities for trogocytotic interactions between microglia

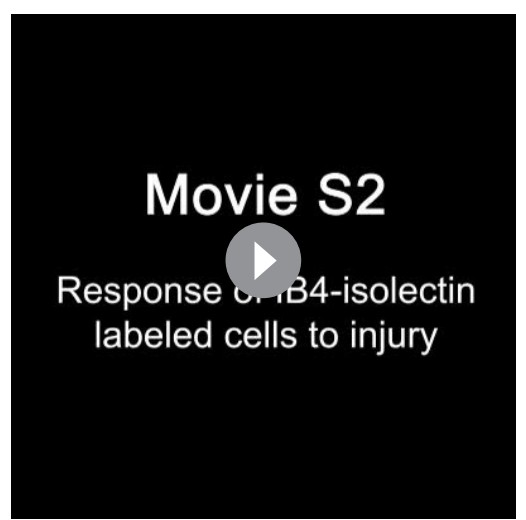

**Video 2.** Response of IB4-isolectin-labeled cells to injury. IB4-isolectin-labeled cells respond to tissue injury by mobilization to the injury site and phagocytosis of injured tissues. Also shown in *Figure 1D*. Timestamp = HH:MM.
https://elifesciences.org/articles/62167#video2

and pHtdGFP-labeled axons, resulting in greater amounts of green fluorescent material found within the microglial population. Based on this principle, we developed an assay to measure microglial trogocytosis of RGC axons in *Xenopus laevis* tadpoles. At developmental stage 39/40, RGC axons were labeled by retinal electroporation with plasmid encoding pHtdGFP, and microglia were labeled by intraventricular injection of Alexa 594-conjugated IB4-isolectin (*Figure 3A*). By 2 days post-labeling, axons begin expressing pHtdGFP and innervate the optic tectum. This also corresponds to the period when microglia begin to extensively colonize the optic tectum. At day 4 and day 5 post-labeling, the optic tectum was imaged by two-photon microscopy. The number of pHtdGFP-labeled axons present in the optic tectum was counted and the green fluorescence within the population of microglia in the imaging field was quantified using 3D masking with the IB4-isolectin channel (*Figure 3B*). To control for the possibility of RGC apoptosis, data were excluded if axonal blebbing was observed, or if the number of axons

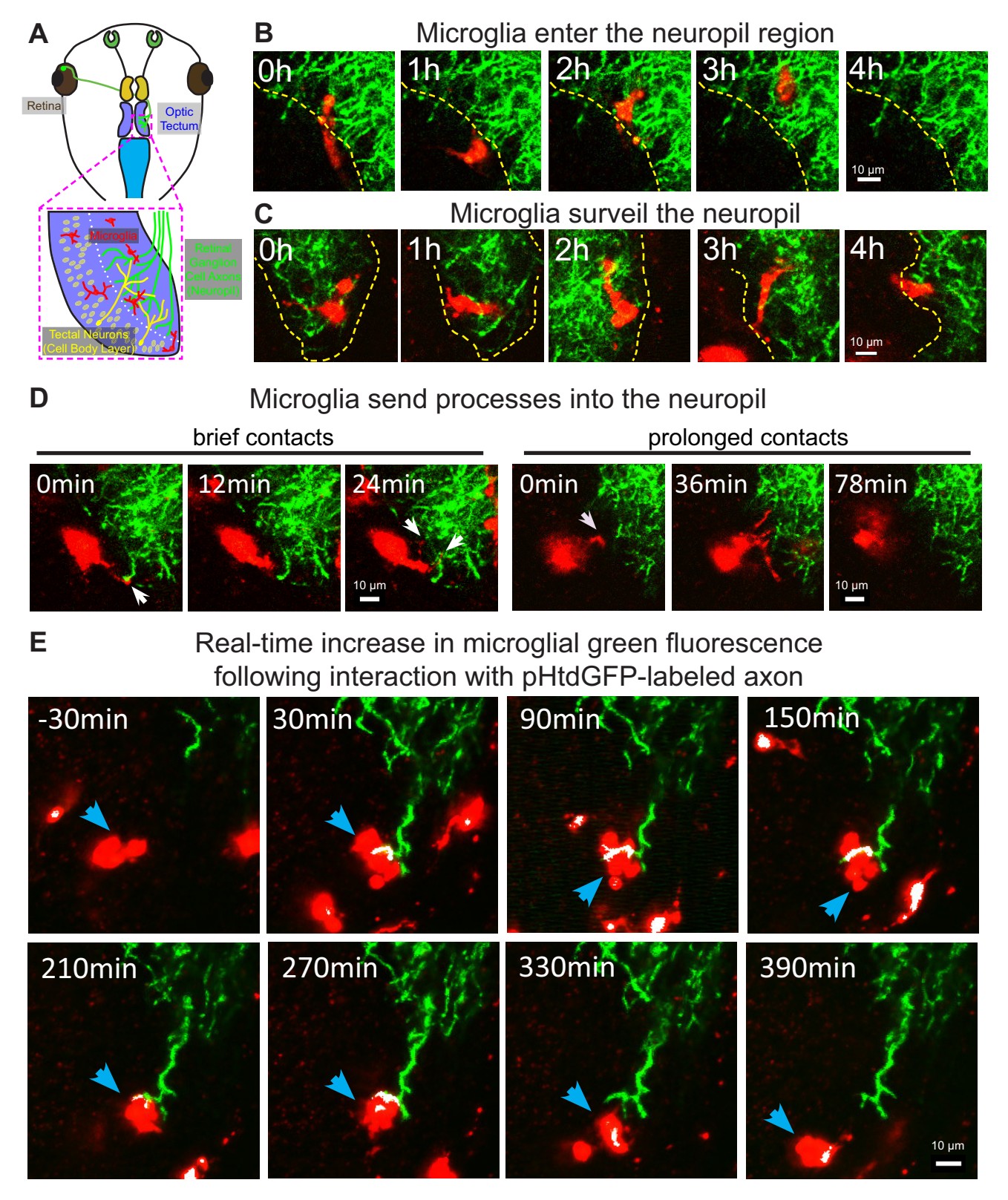

**Figure 2.** Microglia surveil the tectal neuropil, contact RGC axons, and increase in green fluorescence following an interaction with pHtdGFP-labeled axons in real-time. (**A**) A diagram of the developing *Xenopus laevis* retinotectal circuit. (**B**) The tectal neuropil does not exclude microglia in *Xenopus laevis.* The yellow dotted line indicates the border of the cell body layer and the neuropil region. Microglia (red) can be observed migrating in and out

*Figure 2 continued on next page*

*Figure 2 continued*

of the neuropil region from the cell body layer. A single registered optical section is shown. (C) Microglia surveil the neuropil. A microglial cell is followed over time as it traversed different depths in the tectum. (D) Microglia extend processes (white arrows) into the neuropil. Contact duration varied between minutes and hours. (E) A microglial cell (blue arrow) interacts with a pHtdGFP-labeled RGC axon and increases in green fluorescence in real-time. The colocalization of green and red is colorized as white.

The online version of this article includes the following source data and figure supplement(s) for figure 2:

**Figure supplement 1.** Green fluorescence in microglial cells increases in real-time following interaction with pHtdGFP-labeled axons.

**Figure supplement 1—source data 1.** Fluorescence changes in microglial cells in time lapse imaging experiments from *Figure 2—figure supplement 1*.

---

decreased from day 4 to day 5. Even when electroporation yields no labeled axons, some basal green fluorescence in microglial cells is still observed (*Figure 3C* and *Figure 3—figure supplement 1*). This is because microglia have high levels of autofluorescent molecules such as lipofuscin, bilirubin, and porphyrins (*Mitchell et al., 2010*). At day 4, there is a weak positive correlation between microglial green fluorescence and the number of pHtdGFP-labeled axons in the optic tectum, a relationship which is significantly strengthened on day 5 (*Figure 3D*). The change in green fluorescence associated with microglia from day 4 to day 5 significantly correlates with the number of pHtdGFP-labeled axons in the optic tectum (*Figure 3—figure supplement 2A*) suggesting that microglial cells accumulate pHtdGFP from intact axons between day 4 and day 5 post-labeling.

To determine if synaptic material is being trogocytosed by microglia, we generated a synaptophysin-pHtdGFP fusion protein (SYP-pHtdGFP). SYP is a presynaptic vesicle protein (*Valtorta et al., 2004*), and SYP fusion proteins are commonly used as synaptic vesicle markers (*Nakata et al., 1998*; *Ruthazer et al., 2006*). Expressing SYP-pHtdGFP in RGC neurons yielded axons with pHtdGFP puncta along the length of their terminal arbors (*Figure 3E*). When SYP-pHtdGFP is expressed in RGC axons, on day 4, no correlation is observed between microglial green fluorescence and the number of SYP-pHtdGFP-labeled axons in the optic tectum. However, by day 5, a correlation between microglial green fluorescence and the number of SYP-pHtdGFP-labeled axons is observed (*Figure 3F*). The change in microglial green fluorescence from day 4 to day 5 is proportionate to the number of SYP-pHtdGFP-labeled axons in the optic tectum (*Figure 3—figure supplement 2B*), suggesting that microglial cells trogocytose and accumulate presynaptic elements from RGC axons over the period from day 4 to day 5.

## Depletion of microglial cells enhances RGC axon branching and reverses the profile of behavioral responses to dark and bright looming stimuli

To assess the functional roles of microglial trogocytosis, we first depleted microglia using PLX5622. PLX5622 is an inhibitor of colony stimulating factor one receptor (CSF1R), which is a tyrosine kinase receptor essential for microglia survival (*Elmore et al., 2014*; *Erblich et al., 2011*). Animals reared in 10 μM PLX5622 had significantly reduced microglia numbers in the optic tectum compared to vehicle-treated animals (*Figure 4A* and *Figure 4B*). Morphological analysis of surviving microglia also revealed a

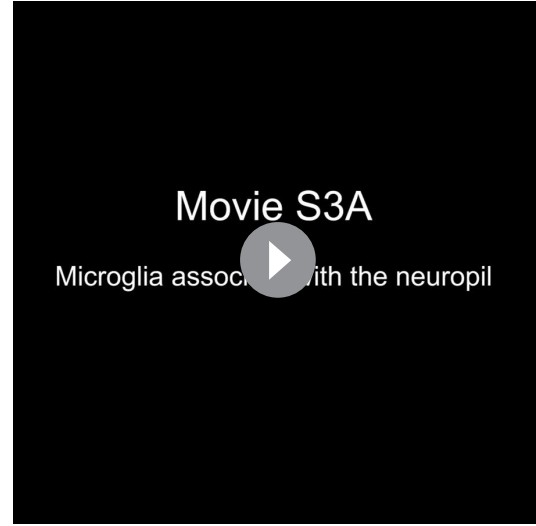

**Video 3.** Microglia surveil the tectal neuropil in developing *Xenopus laevis*. (A) Microglia associate with the tectal neuropil in *Xenopus laevis*. Timestamp = HH: MM. (B) The neuropil does not exclude microglia. Microglia can mobilize into the neuropil region from the cell body layer and can freely move through the neuropil region. Timestamp = HH:MM. (C) Microglia surveil the tectal neuropil by extending processes into the neuropil from the cell body layer. Timestamp = HH: MM.
https://elifesciences.org/articles/62167#video3

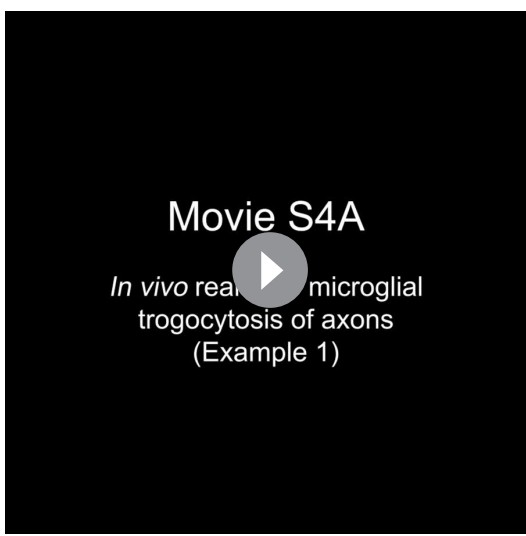

**Video 4.** In vivo real-time trogocytosis in *Xenopus laevis* tadpoles imaged by 2-photon microscopy. (A) A microglial cell increases in green fluorescence in real-time after an interaction with a pHtdGFP-labeled axon. The colocalization of green and red is colorized as white. Timestamp = HH:MM. (B) Another example of a microglial cell increasing in green fluorescence following an interaction with a pHtdGFP-labeled axon. This example is shown in *Figure 2E*. Timestamp = HH: MM.
https://elifesciences.org/articles/62167#video4

reduction in the number of processes per microglial cell, indicating a less ramified morphology (*Figure 4C*).

Next, we interrogated the effect of microglial depletion on the morphology of single RGC axons. Axons were followed for several days in control and microglia-depleted animals (*Figure 4D*). Microglial depletion with PLX5622 did not affect axon arbor length (*Figure 4E*) but significantly increased the number of axon branches (*Figure 4F*), suggesting that microglia negatively regulate axonal arborization during development.

We then sought to delineate the functional effects of microglial depletion on the development of the retinotectal circuit. Previous reports in *Xenopus laevis* and in zebrafish have shown that the retinotectal circuit is a vital processing and decision-making center for the visual detection of looming objects (*Dong et al., 2009*; *Khakhalin et al., 2014*). Therefore, we developed a free-swimming looming stimulus assay in *Xenopus* tadpoles (*Figure 5A*) to determine the functional outcomes of disrupting microglial function in this circuit. Tadpole behavioral responses to dark looming stimuli or bright looming stimuli (*Figure 5B*) were recorded and custom computer vision software was used to track the locomotor response of the tadpole (*Figure 5C*).

*Figure 5D* shows contrails from representative animals in response to dark looming stimuli. In vehicle-treated animals, dark looming stimuli evoked stereotypical defensive escape behavior (*Video 5A*), whereas microglia-depleted animals were less likely to make defensive responses to dark looming stimuli (*Video 5B*). Presentation of dark looming stimuli elicited an increase in velocity in control animals (*Figure 5E*), with a peak in instantaneous velocity at 1.1 ± 0.1 s (mean ± SD, n = 8) post-stimulus. The distance traveled by control tadpoles over the 3 s immediately following presentation of a dark looming stimulus increased compared to the period before the stimulus was presented (*Figure 5F*). This increase was absent in microglia-depleted animals. We categorized tadpole responses to looming stimuli as exhibiting defensive behavior, absence of defensive behavior or undeterminable (excluded from response rate calculation). Microglia-depletion significantly reduced the response rate to dark looming stimuli (*Figure 5G*).

*Figure 5H* shows contrails from representative animals in response to bright looming stimuli. Surprisingly, bright looming stimuli rarely evoked defensive escape behavior in control animals (*Video 6A*) but often evoked robust responses in microglia-depleted animals (*Video 6B*). Presentation of bright looming stimuli elicited an increase in velocity in microglia-depleted animals (*Figure 5I*), with a peak in instantaneous velocity at 1.9 ± 0.6 s (mean ± SD, n = 8) after the onset of stimulus presentation. The distance traveled by microglia-depleted tadpoles over the 3 s immediately following presentation of a bright looming stimulus increased compared to the period before the stimulus was presented (*Figure 5J*). This increase was absent in vehicle-treated animals. Furthermore, microglia-depletion significantly increased the response rate to bright looming stimuli (*Figure 5K*).

In trials where animals exhibited escape behavior to dark or bright looming stimuli, microglia depletion did not significantly alter escape distance, maximum escape velocity, or escape angle, suggesting that microglial depletion did not disrupt motor function (*Figure 5—figure supplement 1*).

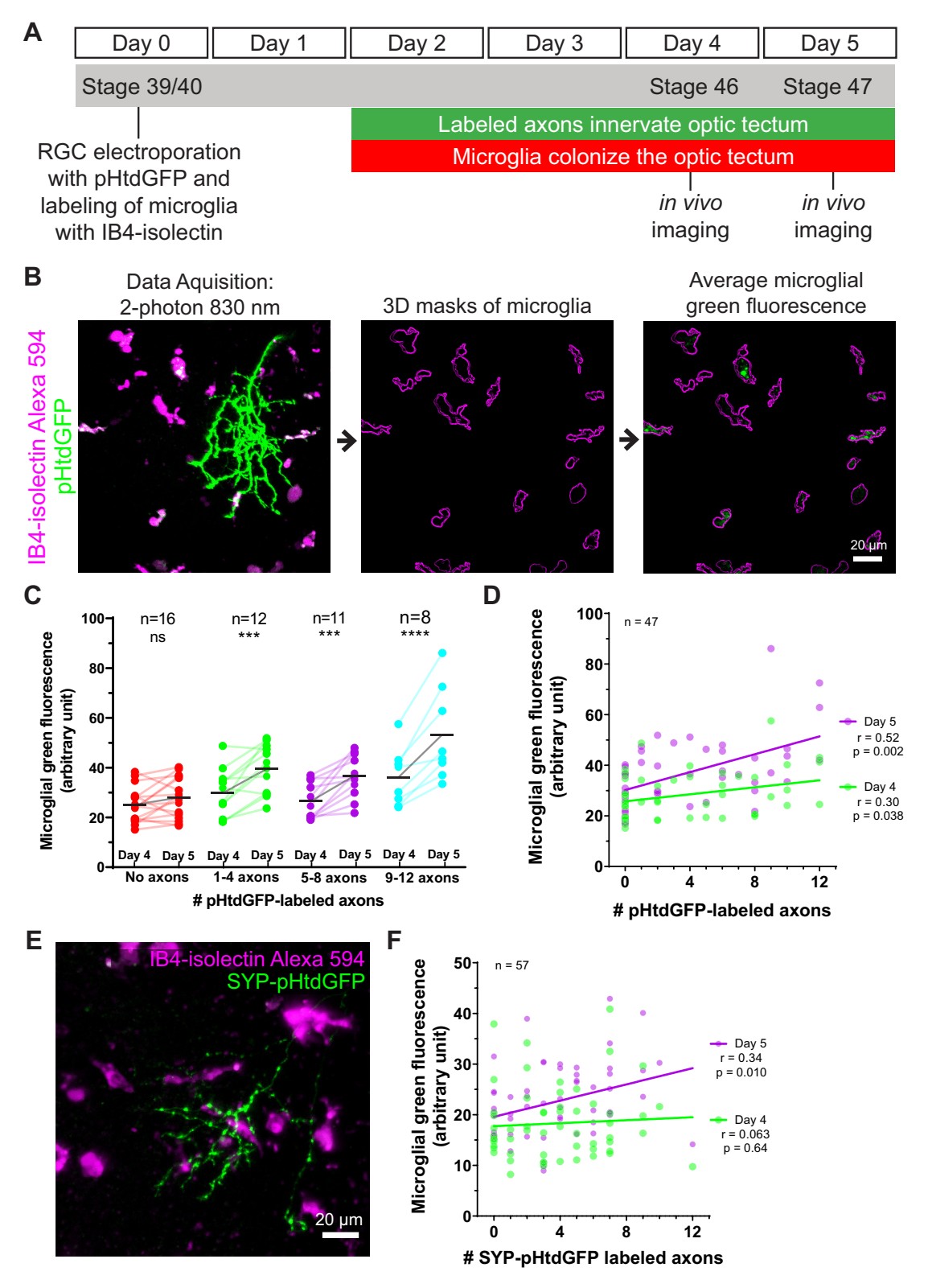

**Figure 3.** Microglia accumulate green fluorescence label from axons expressing pHtdGFP or SYP-pHtdGFP. (**A**) Timeline of trogocytosis assay. Axons were labeled with pH-stable GFP (pHtdGFP). 2-photon imaging was performed on day 4 and day 5 post-labeling. (**B**) Measurement of green fluorescence signal from microglia. pHtdGFP-labeled axons (green) were imaged concurrently with microglia (magenta). 3D microglia ROIs were automatically generated (magenta outlines). The average microglial green fluorescence was quantified from the population of microglia sampled in the

*Figure 3 continued on next page*

*Figure 3 continued*

z-stack. (**C**) Presence of pHtdGFP-labeled axons in the optic tectum increases microglial green fluorescence between day 4 and day 5. Two-way RM ANOVA interaction $F_{(3,43)} = 6.14$, $p=0.0014$. Sidak's multiple comparison test ***$p<0.001$, ****$p<0.0001$. (**D**) Microglial green fluorescence weakly correlates with the number of pHtdGFP-labeled axons in the optic tectum on day 4 (n = 47, Pearson's r = 0.30, p=0.038) and moderately correlates on day 5 (n = 47, Pearson's r = 0.52, p=0.0002). The same dataset is analyzed in **C**. (**E**) SYP-pHtdGFP fusion protein localizes pHtdGFP to presynaptic puncta. (**F**) Microglial green fluorescence does not significantly correlate with the number of SYP-pHtdGFP-labeled axons in the optic tectum on day 4 (n = 57, Pearson's r = 0.063, p=0.64), and weakly correlates on day 5 (n = 57, Pearson's r = 0.34, p=0.010).

The online version of this article includes the following source data and figure supplement(s) for figure 3:

**Source data 1.** Mean microglial-associated green fluorescence as a function of axon number measured over 24 hr.

**Figure supplement 1.** The presence of pHtdGFP-labeled axons significantly increases microglial green fluorescence between day 4 and day 5.

**Figure supplement 1—source data 1.** Histograms of microglial-associated green fluorescence as a function of axon number measured over 24 hr from *Figure 3—figure supplement 1*.

**Figure supplement 2.** Correlation between the change in microglial green fluorescence from day 4 to day 5 and the number of pHtdGFP-labeled and SYP-pHtdGFP-labeled axons.

## Bioinformatic identification of amphibian regulator of complement activation 3 (aRCA3), a neuronally expressed membrane-bound complement inhibitory molecule and homolog of human CD46

Microglial depletion enhanced axonal arborization and perturbed functional development of the retinotectal circuit. As microglia are known to secrete trophic factors and cytokines (*Parkhurst et al., 2013*; *Solek et al., 2018*), signaling mechanisms unrelated to trogocytosis may be at work. To better discern a role of trogocytosis in modulating axon morphology, we searched for a molecule that would inhibit axon trogocytosis. Because complement C3 is a known mediator of phagocytosis (*Brown and Neher, 2014*) and has been proposed to accumulate at synapses and tag them for removal by microglial phagocytosis (*Schafer et al., 2012*; *Stevens et al., 2007*), we carried out a bioinformatics screen to identify neuronally expressed proteins that have potential inhibitory activity against complement C3.

To search for proteins that inhibit complement C3, we first used the STRING Protein-Protein Association Network to identify proteins which interact with human complement C3 (*Figure 6A*). To select for complement inhibitory proteins, our search was then refined by focusing on proteins containing CCP (domains abundant in complement control proteins) motifs, which are highly conserved modules abundant in proteins responsible for negative regulation of the complement system (*Norman et al., 1991*). The top CCP-containing proteins identified by STRING were queried on the Allen Brain Institute human multiple cortical areas RNA-seq dataset (*Allen Institute for Brain Science, 2015*; *Hodge et al., 2019*). Of the CCP-containing proteins identified by STRING, only CD46 is highly expressed by human CNS neurons (*Figure 6B*).

Using NCBI protein-protein BLAST, the *Xenopus laevis* protein with the highest similarity to human CD46 was identified as amphibian Regulator of Complement Activation 3 (aRCA3) (*Table 1*). aRCA3 is a protein uncharacterized in *Xenopus laevis* but characterized as a membrane-associated complement regulatory protein in *Xenopus tropicalis* (*Oshiumi et al., 2009*). Like human CD46, *Xenopus laevis* aRCA3 is a type I transmembrane protein with many extracellular CCP domains and a stop-transfer anchor sequence near the C-terminus of the protein (*Figure 6C*). As a comparison, the second and third most similar *Xenopus laevis* proteins, complement receptor two and complement component four binding protein, do not share a protein architecture with CD46. The tertiary structure of *Xenopus laevis* aRCA3 resembles that of human CD46 (*Video 7A* and *Video 7B*). Despite having twice the number of CCP domains as human CD46, aRCA3 is a likely homolog of CD46. The number of CCP domains of CD46 homologs varies across species—for example, human (*Lublin et al., 1988*) and rodent CD46 have four CCP domains (*Miwa et al., 1998*), the *Gallus gallus* homolog (CREM) has six CCP or CCP-like domains (*Oshiumi et al., 2005*), and homologs in *Danio rerio* (rca2.1) (*Tsujikura et al., 2015*), *Equus caballus* (MCP), and *Coturnix japonica* (MCP-like) each have five CCP domains.

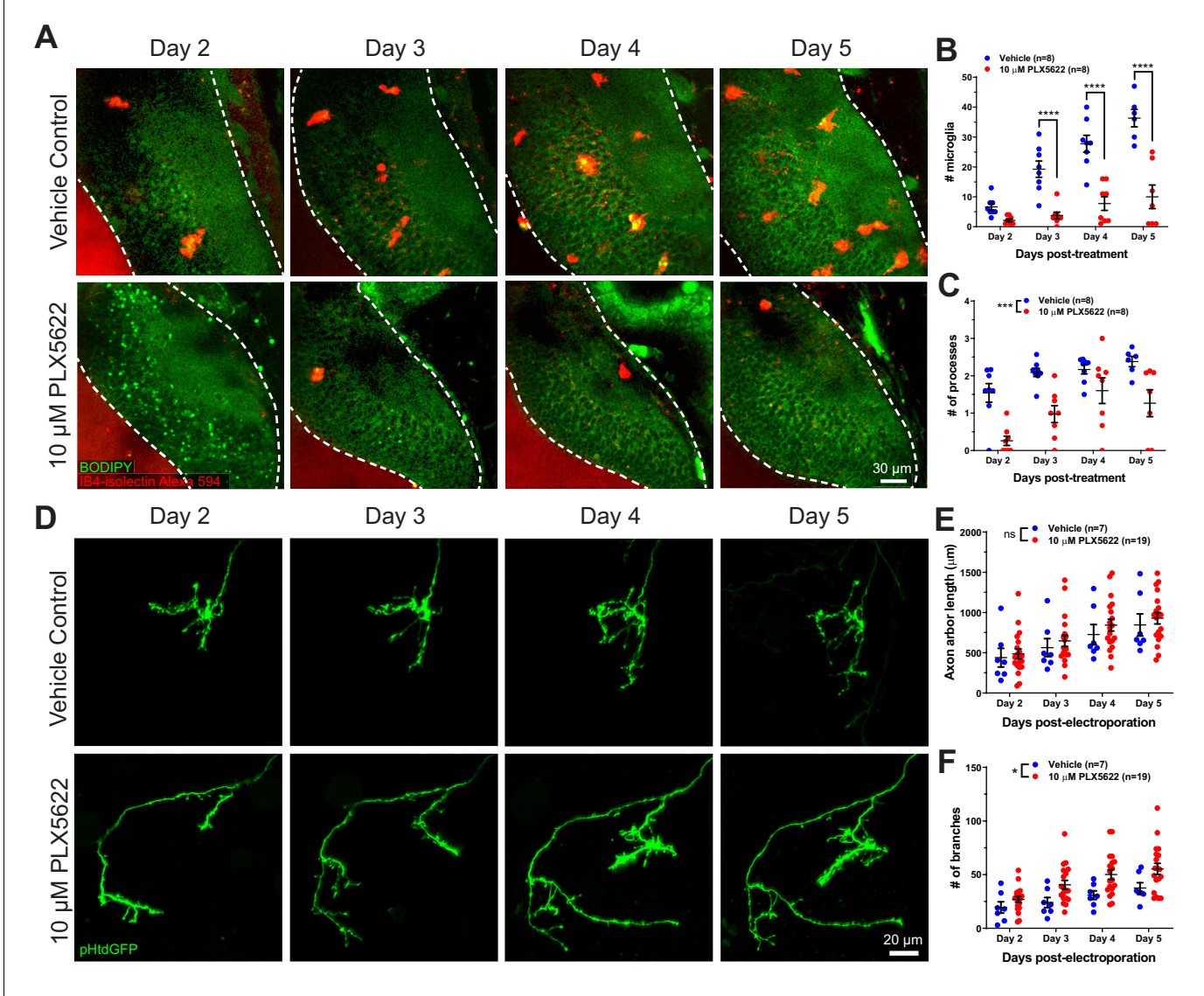

**Figure 4.** CSF1R antagonism depletes microglia from the optic tectum and increases axon arbor branch number. (**A**) Animals were treated with vehicle or 10 μM PLX5622. Brain structures and microglia were labeled using CellTracker Green BODIPY (green) and IB4-isolectin (red), respectively. The white dotted line indicates the border of the optic tectum. Single optical sections are shown. (**B**) PX5622 depletes microglia in the optic tectum. Mixed-effects REML model interaction F(3,39) = 14.23, p<0.0001. Sidak's multiple comparison post-hoc test ****p<0.0001. Data are shown as mean ± SEM. (**C**) PLX5622 reduces the number of processes per microglia. Mixed-effects REML model main effect F(1,14)=18.42, ***p<0.001. Data are shown as mean ± SEM. (**D**) Monitoring of individual RGC axons in vehicle control and PLX5622-treated animals. (**E**) Microglial depletion with PLX5622 did not affect axon arbor length. Two-way RM ANOVA main effect F(1,24)=0.4141, p=0.53. Data are shown as mean ± SEM. (**F**) Microglial depletion with PLX5622 increased axon branch number. Two-way RM ANOVA main effect F(1,24)=5.581, p=0.027. Data are shown as mean ± SEM.

The online version of this article includes the following source data for figure 4:

**Source data 1.** Microglia and axon morphometric properties from time lapse imaging over 4 days.

To investigate whether RGC neurons endogenously express aRCA3, retinal sections were probed by fluorescence RNAscope in situ hybridization (*Wang et al., 2012*). Probes against aRCA3, the positive control transcript RNA polymerase II subunit A (polr2a.L), and the negative control transcript bacterial dihydrodipicolinate reductase (DapB) were hybridized on PFA fixed retinal sections (*Figure 6D*). As expected, DapB transcripts were not detected in the retina, whereas the housekeeping polr2a transcript was ubiquitously expressed. In contrast, aRCA3 transcripts localize to the

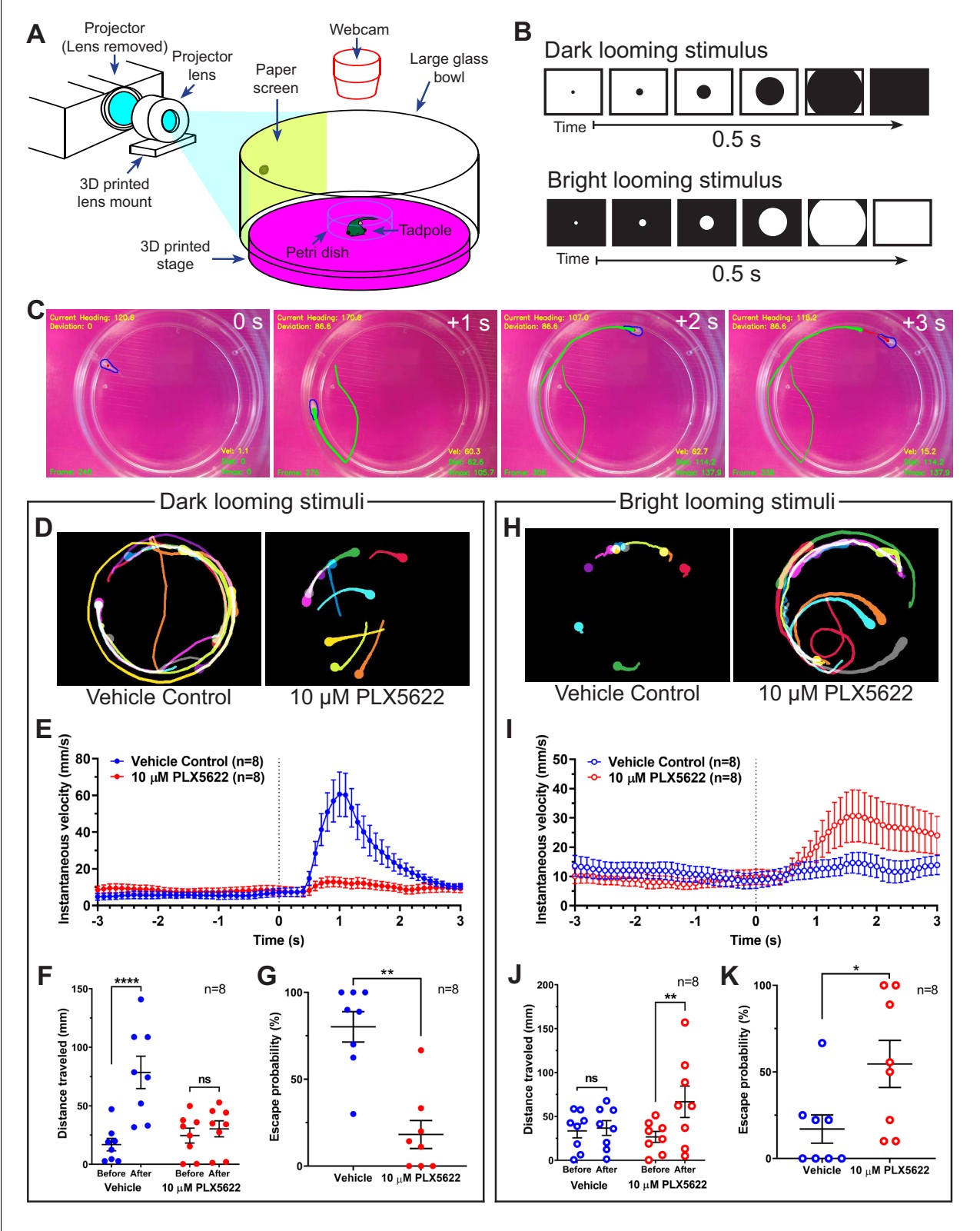

**Figure 5.** Microglial depletion reverses the expected behavioral response to both dark and bright looming stimuli. (A) Schematic of a looming behavioral task to assess visuomotor responses in *Xenopus laevis* tadpoles. Stage 47 animals were presented looming stimuli and free-swimming escape responses were recorded. (B) Exponentially expanding dark and bright circles were presented as looming stimuli. (C) Representative response to dark looming stimulus (presented at 0 s) in a vehicle-treated animal. A contrail is drawn from 0 to 2 s post-stimulus. (D) Representative contrails of

Figure 5 continued

the escape responses to dark looming stimuli in a single animal (10 trials). (**E**) After the dark looming stimulus is presented (0 s), vehicle-treated animals (blue) increase in velocity, whereas microglia-depleted animals (red) do not. n = 8. Data are shown as mean ± SEM. (**F**) Presentation of dark looming stimuli increases distance traveled over 3 s in vehicle-treated animals (blue) but not microglia-depleted animals (red). 2-way RM ANOVA interaction F (1,14) = 15.82, p=0.0014. Sidak's multiple comparisons test ****p<0.0001. Data are shown as mean ± SEM. (**G**) Microglia depletion reduces the escape probability to dark looming stimuli. Mann-Whitney test **p<0.01, n = 8. Data are shown as mean ± SEM. (**H**) Representative contrails to bright looming stimuli. (**I**) After the bright looming stimulus is presented (0 s), microglia-depleted animals (red) increase in velocity, whereas vehicle-treated animals (blue) do not. n = 8. Data are shown as mean ± SEM. (**J**) Presentation of bright looming stimuli increases distance traveled over 3 s in microglia-depleted animals (red) but not vehicle-treated animals (blue). Two-way RM ANOVA interaction F(1,14) = 5.291, p=0.037. Sidak's multiple comparisons test **p<0.01. Data are shown as mean ± SEM. (**K**) Microglia depletion increases the escape probability to bright looming stimuli. Mann-Whitney test *p<0.05, n = 8. Data are shown as mean ± SEM.

The online version of this article includes the following source data and figure supplement(s) for figure 5:

**Source data 1.** Effects of microglial depletion on escape behavior induced by looming stimuli.

**Figure supplement 1.** Motor characteristics of escape behavior are not significantly altered by PLX5622-induced microglial depletion.

ganglion cell layer (*Figure 6E*), revealing that the aRCA3 gene is endogenously expressed in RGC neurons.

## aRCA3 associates with synapses, enhances axonal arborization and inhibits microglial trogocytosis

It is unknown whether aRCA3 localizes to the correct cellular compartments to protect axons from complement attack. To investigate this, we examined the localization of aRCA3 by tagging it with mCherry (*Figure 7A*). This aRCA3-mCherry fusion protein was expressed together with SYP-pHtdGFP, which localizes to synaptic puncta (*Figure 3E*). SYP-pHtdGFP and aRCA3-mCherry colocalize when co-expressed in the same axon (*Figure 7B*). The high degree of colocalization between SYP-pHtdGFP and aRCA3-mCherry suggests that aRCA3 distributes to similar subcellular compartments as the presynaptic marker SYP (*Figure 7C*).

Next, we examined the effect of aRCA3 overexpression on microglial trogocytosis. pHtdGFP was expressed in RGC neurons with or without overexpression of aRCA3, and microglial trogocytosis was quantified by measuring microglial green fluorescence on day 4 and day 5 post-labeling. Expression of pHtdGFP in RGC neurons resulted in a significant correlation between microglial green fluorescence and the number of pHtdGFP-labeled axons on day 5 post-labeling (*Figure 7D*), a relationship that was not observed when aRCA3 was overexpressed (*Figure 7E*). Overexpression of aRCA3 significantly reduced the correlation coefficient between the change in microglial green fluorescence from day 4 to day 5 and the number of pHtdGFP-labeled axons (*Figure 7—figure supplement 1*), suggesting that overexpression of aRCA3 reduces microglial trogocytosis.

We then examined the effect of aRCA3 overexpression on axon morphology in single RGC axons over several days (*Figure 7F*). Overexpression of aRCA3 resulted in significantly greater axon arbor length compared to control axons (*Figure 7G*). Similarly, axon branch number was greater in axons overexpressing aRCA3 compared to control axons (*Figure 7H*). These results show that aRCA3 overexpression causes axons to have larger and more highly branched arbors.

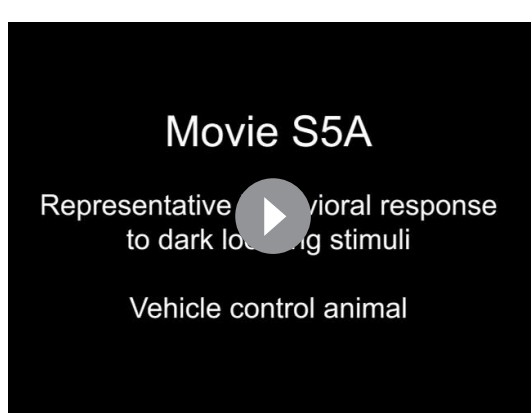

**Video 5.** Representative responses to dark looming stimuli in control and microglia-depleted animals. (A) Representative response to dark looming stimuli in a vehicle control animal. (B) Representative response to dark looming stimuli in a microglia-depleted animal.
https://elifesciences.org/articles/62167#video5

While CD46 is best known for its ability to inactivate complement C3 and complement C4, CD46 can also signal through intracellular tyrosine kinase activity under certain conditions (*Riley-Vargas et al., 2004*). This raises the potential caveat that the increased axonal arborization induced by overexpression of aRCA3 may result from aberrant intracellular signaling. Using a bioinformatic tool, NetPhos3.1 with cutoff scores > 0.6 (*Blom et al., 2004*), we did not find a predicted tyrosine kinase phosphorylation site on the cytoplasmic region of aRCA3. Nonetheless, it is possible that aRCA3 may exert some of its effects through an intracellular signaling pathway. To provide further functional validation that the complement system affects axonal arborization, we examined the effects of enhancing complement activity on axon morphology.

**Video 6.** Representative responses to bright looming stimuli in control and microglia-depleted animals. (A) Representative response to bright looming stimuli in a vehicle control animal. (B) Representative response to bright looming stimuli in a microglia-depleted animal.
https://elifesciences.org/articles/62167#video6

## Expression of a membrane-bound complement C3 fusion protein reduces RGC axon size and branching

If aRCA3 affects axon morphology by inhibiting complement activity, enhancing complement activity on single axons should produce effects opposite to that of aRCA3 overexpression. To explore this possibility, we designed an axon surface-localized complement C3 fusion protein to enhance complement activity on individual axons (*Figure 8A*). Synaptobrevin, also known as vesicle-associated membrane protein 2 (VAMP2), is concentrated in synaptic vesicles, though a significant fraction of VAMP2 is also present on the axon surface (*Ahmari et al., 2000*; *Sankaranarayanan and Ryan, 2000*). We cloned *Xenopus laevis* complement C3 and fused the N-terminus to the extracellular C-terminus of *Xenopus laevis* VAMP2. This design was chosen in favor of a GPI anchor design that modifies the C-terminus C345C domain of complement C3 as this domain undergoes major rearrangement during activation and proteolysis (32° rotation, 10 Å translation) (*Janssen et al., 2005*). In contrast, the N-terminus of complement C3 is exposed on the surface of the protein and located on the MG1 domain, a domain that does not undergo marked confirmational changes upon complement C3 activation and proteolysis (3° rotation, 1 Å translation). Thus, expression of VAMP2-C3 in RGC neurons results in axons tagged with extracellular membrane-bound complement C3. While we used complement C3 precursor to generate the VAMP2-C3 fusion protein, complement C3 undergoes spontaneous, low-level activation through the alternative complement pathway (*Pangburn et al., 1981*).

VAMP2-C3 was co-expressed with pHtdGFP in RGC neurons, and axons were monitored over several days (*Figure 8B*). To control for the possibility that VAMP2 overexpression may affect axon morphology, we also overexpressed VAMP2 alone in RGC axons. Expression of VAMP2-C3 in RGC axons significantly reduced axon arbor length and axon branch number when compared to control or VAMP2 overexpression (*Figure 8C* and *Figure 8D*), demonstrating that enhancing complement activity negatively regulates axonal arborization.

## Discussion

### In vivo evidence of microglial trogocytosis of axons during healthy development

Previous real-time imaging experiments in ex vivo slice culture have shown that microglia trogocytose presynaptic elements (*Weinhard et al., 2018*). Our real-time imaging results now add in vivo support to the hypothesis that microglia engulf synaptic material via trogocytosis. Additionally, labeling of axons with pHtdGFP or SYP-pHtdGFP resulted in an increase in fluorescent label within

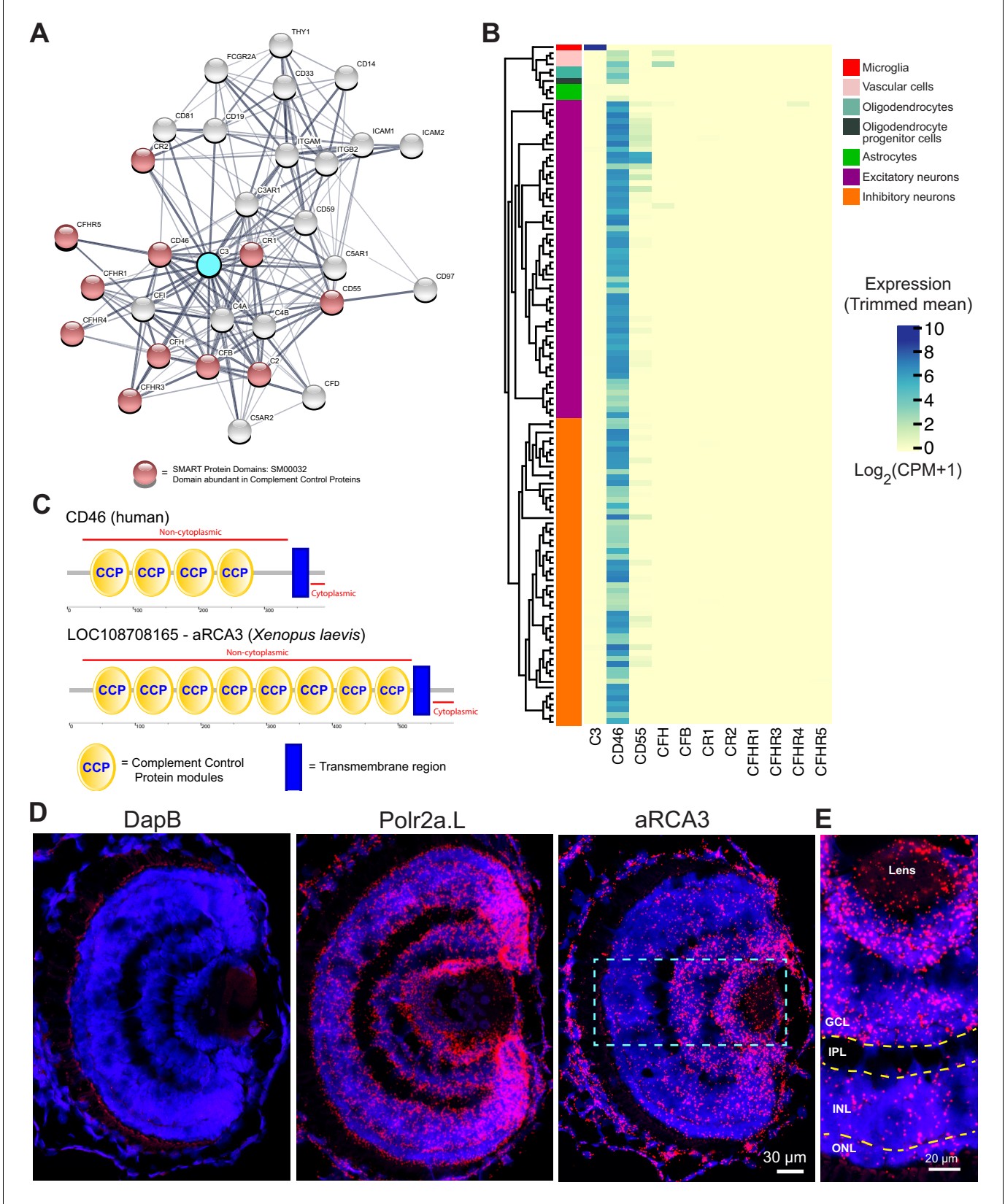

**Figure 6.** Identification of a neuronally expressed membrane-bound complement inhibitory protein, amphibian regulator of complement activation 3 (aRCA3), the predicted homolog of human CD46. (**A**) Complement C3 was queried on the STRING Protein-Protein Association Network and the top 30

*Figure 6 continued on next page*

*Figure 6 continued*

interaction partners are displayed. Red nodes represent proteins which contain complement control domains that inhibit complement activity. Line thickness indicates the strength of data supporting protein interaction. (B) Complement inhibitory proteins identified by STRING were screened for neuronal expression in the Allen Brain Map human cortical transcriptomics dataset. Cell type taxonomy and hierarchical clustering was determined according to previous analysis (*Hodge et al., 2019*). Only CD46 is highly expressed by neurons. Heat map color scale denotes log two expression levels as represented by trimmed mean (25–75%). CPM = counts per million. (C) Protein architecture of CD46 and the most similar *Xenopus laevis* homolog (aRCA3). Both human CD46 and *Xenopus laevis* aRCA3 are type I transmembrane proteins, with a non-cytoplasmic region that contains many complement control protein modules, and a transmembrane anchor near the C-terminus. (D) Fluorescent RNAscope in situ hybridization on retina sections shows that aRCA3 is endogenously expressed in the retina. Negative control probe DapB was not detected in the retina. Housekeeping gene Polr2a.L was expressed ubiquitously. (E) High magnification of the highlighted region in *Figure 5D*. aRCA3 is highly expressed in the GCL, and is present at lower levels in the INL and ONL. GCL = Ganglion Cell Layer; IPL = Inner Plexiform Layer; INL = Inner Nuclear Layer; ONL = Outer Nuclear Layer.

The online version of this article includes the following source data for figure 6:

**Source data 1.** Human cell type expression profiles from the Allen Human Brain database for probable complement inhibitory proteins identified by STRING.

---

microglia between days 4 and 5 post-labeling. As SYP-pHtdGFP is primarily localized to synaptic vesicles, this suggests that the axonal material that microglia engulf contains presynaptic vesicles and is consistent with correlative electron microscopy studies that have demonstrated putative presynaptic vesicles within microglia (*Weinhard et al., 2018*), although we did not use correlative electron microscopy to confirm this in our study.

Numerous past studies have shown that during development microglia have synaptic components within phagocytic compartments (*Paolicelli et al., 2011*; *Schafer et al., 2012*; *Stevens et al., 2007*; *Tremblay et al., 2010*). However, apoptosis is a prominent feature of neural development (*Nijhawan et al., 2000*), and it is unclear whether the source of synaptic material within microglia is because of engulfment of apoptotic neuronal components or if it is due to engulfment of synaptic material from live neurons. When investigating trogocytosis, we minimized the effect of apoptosis by excluding measurements when the number of axons between imaging sessions decreased or when axonal blebbing was observed, providing in vivo direct evidence that microglia accumulate fluorescent label from living axons.

## Expression of complement inhibitory or complement enhancing molecules in neurons regulates axon morphology and trogocytosis by microglia

Activation of complement C3 exposes a reactive thioester bond that covalently attaches to amine or carbohydrate groups on cell surfaces (*Sahu et al., 1994*). Microglia express CR3, a receptor for complement C3 (*Ling et al., 1990*), and binding of CR3 to its ligand induces phagocytosis (*Newman et al., 1985*). Complement C3 localizes to synapses during development and tags synapses for removal by microglia (*Stevens et al., 2007*). The importance of complement C3 in synaptic pruning has been studied by disrupting the C3 pathway using C3 KO mice (*Stevens et al., 2007*), CR3 KO mice (*Schafer et al., 2012*), or recently, with the exogenous neuronal expression of the complement inhibitory proteins Crry (*Werneburg et al., 2020*) and CD55 (*Wang et al., 2020*). Here we show that enhancing the C3 pathway with a membrane-bound VAMP2-C3 fusion protein increases axonal pruning at the single axon level. Conversely, overexpression of an endogenous

---

**Table 1.** Top three most similar proteins to human CD46 in the *Xenopus laevis* genome.

| | Protein name | Query coverage | Percent identity | E-value |
|---|---|---|---|---|
| 1 | Amphibian Regulator of Complement Activation 3 (aRCA3) | 64% | 34.77% | three $\times$ 10$^{-41}$ |
| 2 | Complement Receptor Type 2 | 70% | 31.77% | four $\times$ 10$^{-35}$ |
| 3 | Complement Component 4 Binding Protein | 76% | 31.17% | four $\times$ 10$^{-35}$ |

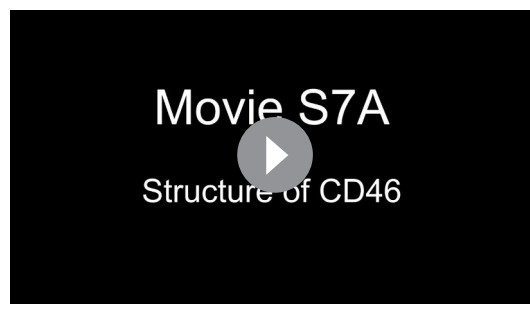

**Video 7.** Predicted tertiary protein structure of CD46 and aRCA3. (A) The predicted tertiary protein structure of CD46. (B) The predicted tertiary protein structure of aRCA3.
https://elifesciences.org/articles/62167#video7

membrane-bound complement inhibitory molecule inhibits axonal pruning and trogocytosis (*Figure 9A*).

While VAMP2-C3 clearly produced effects on axon morphology, we did not rule out the possibility that the expression of VAMP2-C3 in RGC neurons may be producing effects on axons that are not mediated by the complement system. Additionally, we were unable to determine whether VAMP2-C3 expression enhanced microglial trogocytosis as electroporation of RGC neurons with VAMP2-C3 rarely yielded labeled axons. We speculate that expression of VAMP2-C3 in RGC neurons reduces viability, as previous studies have shown that the complement cascade mediates phagocytosis of RGC neurons during development (*Anderson et al., 2019*). Furthermore, our study indirectly examined the effects of VAMP2-C3 and aRCA3 on microglial trogocytosis. Future experiments could directly examine the effects of VAMP2-C3 expression and aRCA3 overexpression on trogocytosis rates by real-time imaging.

It is hypothesized that neurons endogenously express membrane-bound complement inhibitory molecules to protect synapses from phagocytosis (*Stephan et al., 2012*; *Stevens et al., 2007*), and our data support this hypothesis. Recently, sushi repeat protein X-linked 2 (SRPX2) has been identified as a endogenous neuronal complement inhibitor in the mammalian system which protects synapses from complement C1 tagging (*Cong et al., 2020*). SRPX2 is a secreted protein, and while secreted complement inhibitory molecules can act via cell-autonomous mechanisms to protect the neuron that produced it, such molecules also have non-autonomous actions due to the ability to diffuse through the extracellular environment. Conversely, a membrane-bound complement inhibitory molecule acts solely through cell-autonomous mechanisms and allows for greater spatiotemporal control of local complement protection. While the identity of a membrane-bound complement inhibitory molecule endogenous to mammalian neurons remains elusive, in this study we characterized *Xenopus laevis* aRCA3, an endogenously expressed, synaptic vesicle-associated, complement inhibitory molecule. aRCA3 is the most similar amphibian homolog of human CD46. CD46 is a membrane-bound complement inhibitory molecule that cleaves activated complement C3 and C4 (*Barilla-LaBarca et al., 2002*). Using the Allen Brain Institute human multiple cortical areas RNA-seq dataset (*Allen Institute for Brain Science, 2015*), we report that CD46 transcripts are enriched in human neurons. Interestingly, CD46 associates directly with β1-integrins (*Lozahic et al., 2000*), an adhesion molecule present on neuronal surfaces (*Neugebauer and Reichardt, 1991*) and enriched in synaptosomes (*Chan et al., 2003*), suggesting that it too may be localized to axon surfaces and synapses. In our study, we show that aRCA3 expression inhibits both axon trogocytosis by microglia and axonal pruning—we speculate that CD46 may perform similar functions in mammals. Excessive synaptic pruning is thought to be one of the underlying causes of schizophrenia (*Sellgren et al., 2019*), and three large-scale genetic susceptibility studies have identified the CD46 gene as a significant Schizophrenia-risk locus (*Håvik et al., 2011*; *Kim et al., 2020*; *Ripke et al., 2014*). Clearly, the role of CD46 in neurodevelopment warrants further study.

## Microglia actively suppress exuberant arborization at the single axon level

Disrupting microglial function by depletion increases axon tract innervation in prenatal models (*Pont-Lezica et al., 2014*; *Squarzoni et al., 2014*). As depleting microglia increases the number of neural progenitor cells (*Cunningham et al., 2013*) and the number of RGC neurons in the embryonic retina (*Anderson et al., 2019*), it was unclear whether the increased axon tract innervation that occurs following microglial depletion resulted from a deficit of microglial-mediated axonal pruning, or whether it was because of an increase in the overall number of axons. To address this, we examined the effect of microglial depletion on the morphology of single axons, showing that microglia

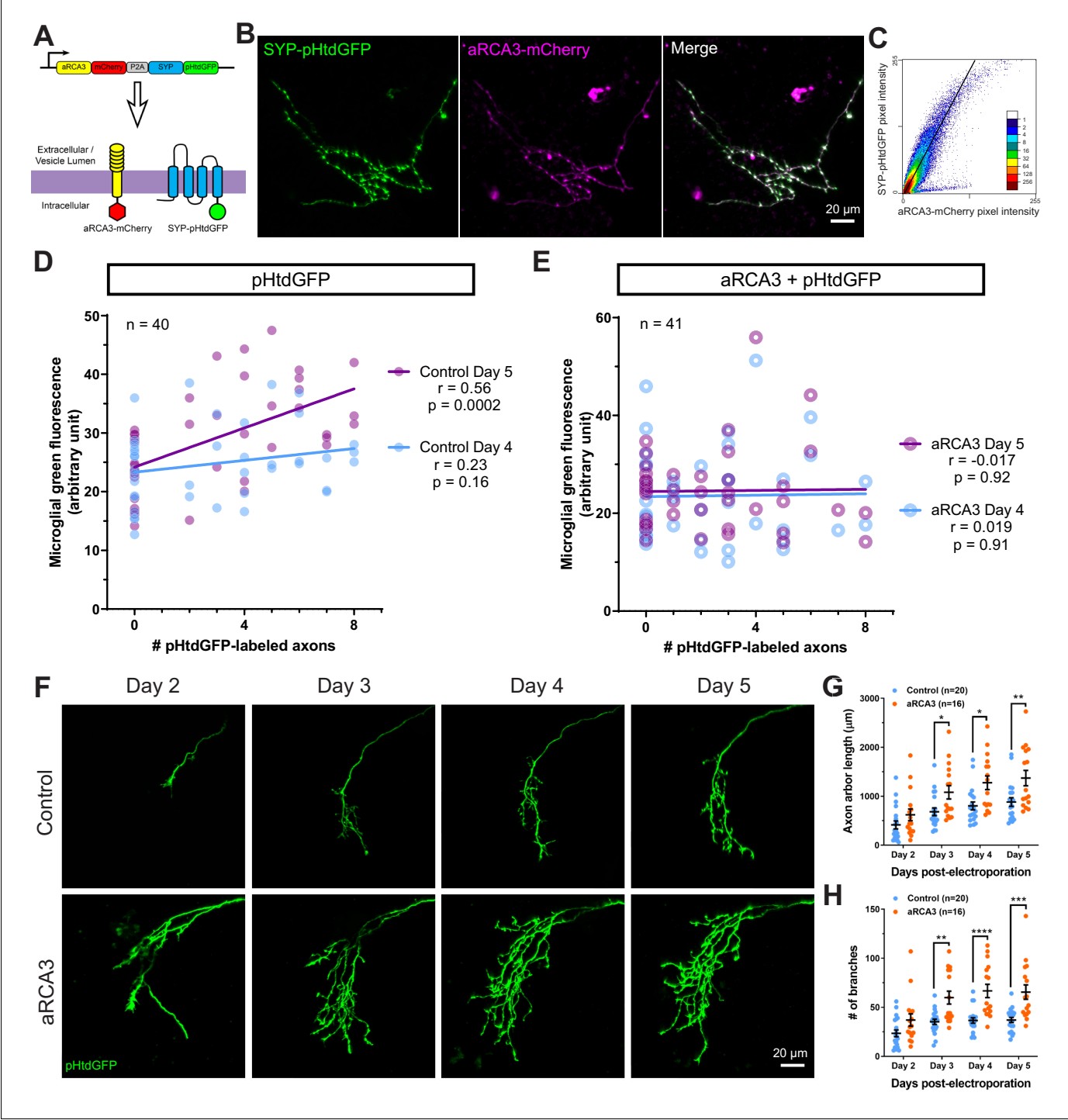

**Figure 7.** RGC overexpression of synapse-associated aRCA3 inhibits microglial trogocytosis and increases axon arborization. (**A**) An aRCA3-mCherry fusion protein was co-expressed with SYP-pHtdGFP in RGC axons. (**B**) Representative RGC axon expressing aRCA3-mCherry and SYP-pHtdGFP. SYP-pHtdGFP is concentrated at synaptic puncta. aRCA3-mCherry colocalizes with SYP-pHtdGFP. (**C**) Colocalization analysis of the axon shown in B. Scatterplot of pixel intensities shows a high degree of association between aRCA3-mCherry and SYP-pHtdGFP. After applying Costes threshold, Pearson's r = 0.90. (**D**) Microglial green fluorescence is not significantly correlated with the number of pHtdGFP-labeled axons in the optic tectum on day 4 (n = 40, Pearson's r = 0.23, p=0.16), and moderately correlated on day 5 (n = 40, Pearson's r = 0.56, p=0.0002). (**E**) When aRCA3 is overexpressed, microglial green fluorescence is not significantly correlated with the number of pHtdGFP-labeled axons in the optic tectum on either day 4 (n = 41, Pearson's r = 0.019, p=0.91) or on day 5 (n = 41, Pearson's r = 0.017, p=0.92). (**F**) Individual control and aRCA3 overexpressing RGC axons were imaged over several days. (**G**) aRCA3 overexpression increases axon arbor length. Two-way RM ANOVA interaction $F_{(3,102)}$ = 5.43, p=0.0017. Sidak's multiple

*Figure 7 continued on next page*

*Figure 7 continued*

comparison test *p<0.05, **p<0.01. Data are shown as mean ± SEM. (**H**) aRCA3 overexpression increases axon branch number. Two-way RM ANOVA interaction F(3,102) = 4.73, p=0.0039. Sidak's multiple comparison test **p<0.01, ***p<0.001, ****p<0.0001. Data are shown as mean ± SEM.

The online version of this article includes the following source data and figure supplement(s) for figure 7:

**Source data 1.** Colocalization analysis for aRCA3-mCherry and SYP-pHtdGFP.

**Source data 2.** Changes in green fluorescence associated with microglia and axonal morphometric analysis comparing control and aRCA3 overexpressing axons.

**Figure supplement 1.** Correlation between the change in microglial green fluorescence from day 4 to day 5 and the number of pHtdGFP-labeled axons with and without aRCA3 overexpression.

depletion significantly increased axon branch number. Thus, our study supports the model that the increase in axon tract innervation resulting from microglial depletion is because of increased arborization of individual axons (*Figure 9B*).

The observations that microglial depletion and overexpression of the complement inhibitor aRCA3 in RGC neurons enhanced axonal arborization suggests that both phenomena may act through a common mechanism. These results support the hypothesis that microglial trogocytosis

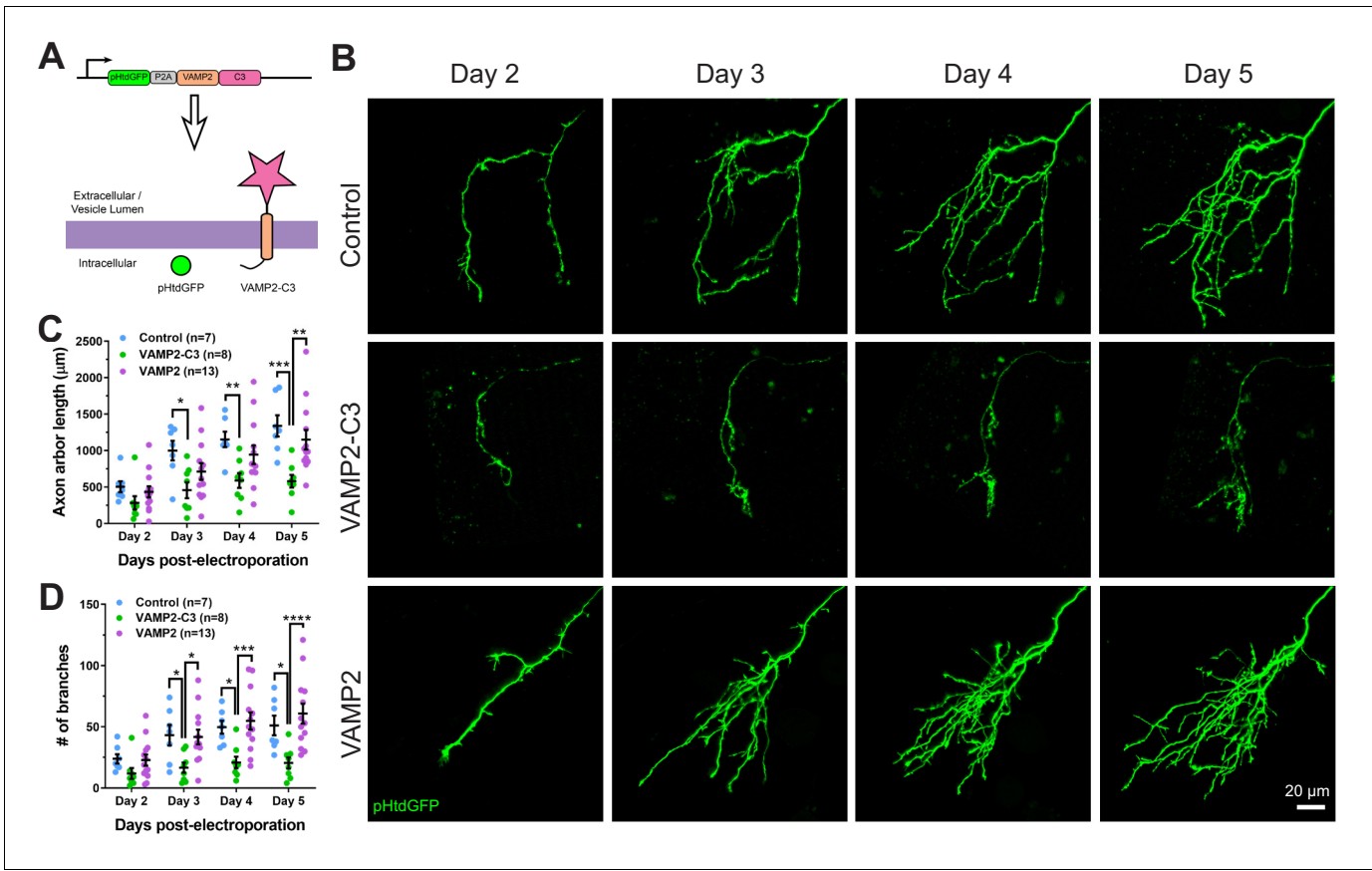

**Figure 8.** Expressing VAMP2-C3 fusion protein in RGC neurons reduces axon arbor size and branch number. (**A**) Complement C3 was fused to the C-terminus of VAMP2 to create a complement enhancing molecule which is targeted to the axon surface and synapses. (**B**) Axons from control, VAMP2-C3 expressing, and VAMP2 overexpressing RGC neurons were imaged over several days. (**C**) Expression of VAMP2-C3 reduces RGC axon arbor length compared to control and VAMP2 overexpression. Two-way RM ANOVA interaction F(6,75)=3.48, p=0.0044. Sidak's multiple comparison test *p<0.05, **p<0.01, ***p<0.001. Data are shown as mean ± SEM. (**D**) Expression of VAMP2-C3 reduces RGC branch number compared to control and VAMP2 overexpression. Two-way RM ANOVA interaction F(6,75)=4.15, p=0.0012. Sidak's multiple comparison test *p<0.05, ***p<0.001, ****p<0.0001. Data are shown as mean ± SEM.

The online version of this article includes the following source data for figure 8:

**Source data 1.** Morphometric analyses comparing control, VAMP2-C3 and VAMP2 expressing axons over 4 days.

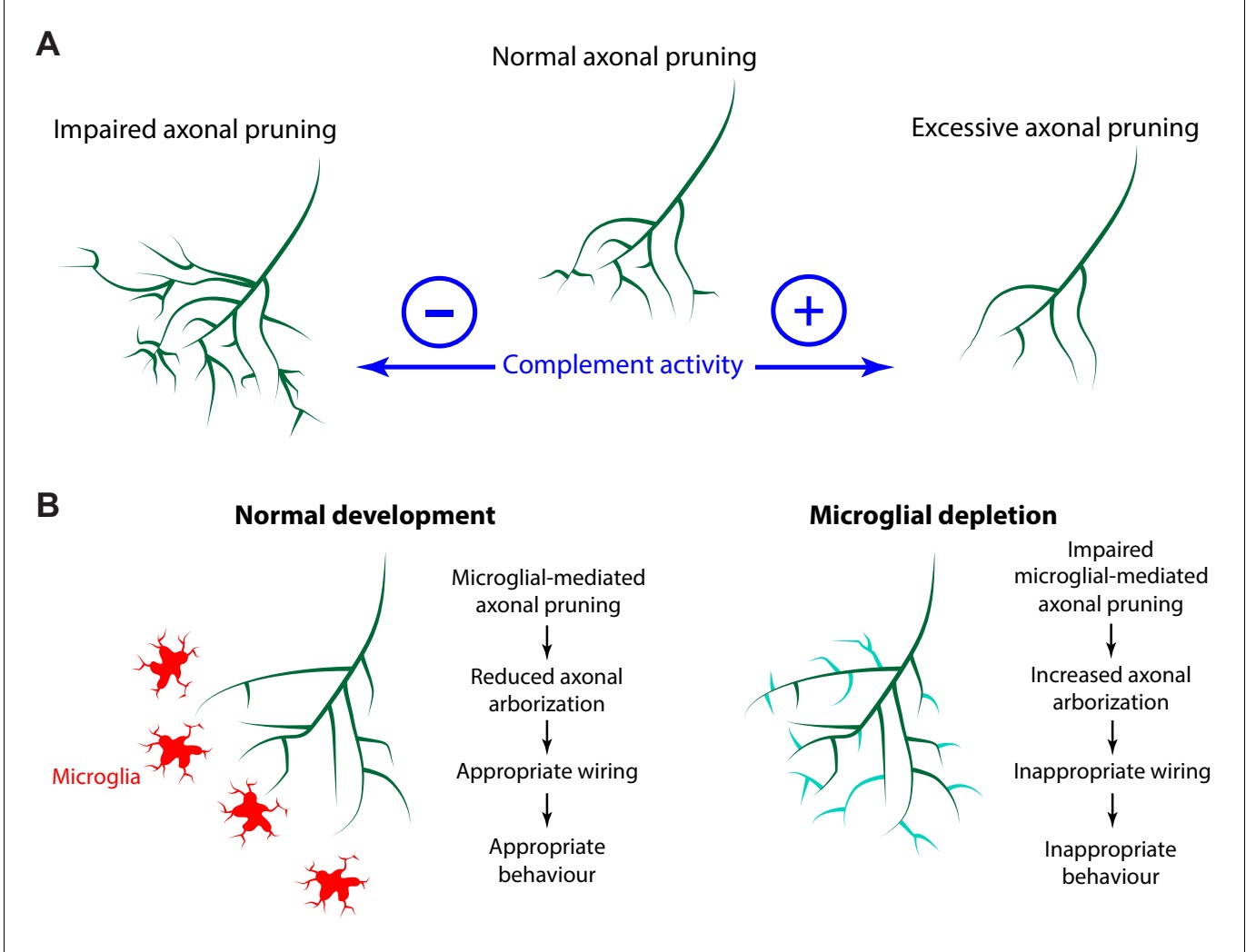

**Figure 9.** The complement system and microglia regulate axonal pruning at the single axon level. (A) Inhibiting complement activity through overexpression of a membrane-bound complement inhibitory molecule results in impaired axonal pruning; conversely, increasing complement activity through expression of a membrane-bound complement enhancing molecule results in excessive axonal pruning. (B) During normal development, microglia actively trogocytose and prune axons. Disrupting microglial function via microglial depletion increases axonal arborization and also disrupts the proper response to dark and bright looming stimuli.

suppresses both axon branching and axon growth through the removal of extra branches and axonal material. While this explanation is at odds with the observation that microglia-depletion did not increase axon arbor length, it is important to note that with aRCA3 overexpression, a single axon gains a relative growth advantage over other axons—this is distinct from the case of microglial depletion, where all axons profit from the same increased growth. If all axons are growing larger at the same pace, competitive mechanisms (*Gosse et al., 2008*; *Ruthazer et al., 2003*) might be expected to constrain axon arbor sizes. Another interesting possibility is that microglia preferentially engage in trogocytosis at short, dynamic branches over longer, established, mature branches. In this case, microglial depletion results in a greater number of short branches, which over the short time course in this study, do not significantly impact axon branch length.

It is worth considering that there may be unintended side-effects of microglial depletion with PLX5622 and that many microglial functions beyond trogocytosis of axons may be altered. In our

study, we found that CSF1R inhibition reduced process branching of remaining microglial cells, an observation that has also been seen in rodents (*Elmore et al., 2014*). Decreased microglial process complexity often accompanies release of cytokines and other neuromodulatory molecules (*Karperien et al., 2013*; *Hanisch, 2002*). While we did not study the effect of microglial depletion on cytokine and chemokine production in this study, past studies have shown that cytokines and chemokines in non-disease mouse models are not altered by PLX5622 (*Reshef et al., 2017*). A further potential confound is that low expression of CSF1R has been reported in some neurons in mice hippocampus (*Luo et al., 2013*). It is possible that CSF1R is present in *Xenopus laevis* RGC neurons and we did not exclude the possibility that PLX5622 may be acting directly on RGC axons in our study.

## Microglia contribute to proper wiring of the retinotectal circuit in developing *Xenopus laevis*

A feed-forward network drives visually evoked escape behavior in *Xenopus* tadpoles (*Khakhalin et al., 2014*). In the retina, photoreceptors act on bipolar cells, which act on RGCs. Some RGCs are tuned to detect looming (*Dunn et al., 2016*; *Münch et al., 2009*), and the population of RGCs that respond to dark looming stimuli are distinct from the population of RGCs that respond to bright looming stimuli (*Temizer et al., 2015*). In zebrafish, these distinct RGC populations project to different arborization fields in the optic tectum (*Robles et al., 2014*; *Temizer et al., 2015*) where the execution of looming computation and behavioral decision-making occurs (*Barker and Baier, 2015*; *Fotowat and Gabbiani, 2011*). In highly predated animals such as crabs, zebrafish, and mice (*Oliva et al., 2007*; *Temizer et al., 2015*; *Yilmaz and Meister, 2013*), dark looming stimuli—which signal imminent threat such as an oncoming object or predator—reliably induce escape responses, whereas bright looming stimuli—which may occur as the animal exits a tunnel or traverses its environment—are less effective.

Microglial depletion does not appear to alter motor functionality, suggesting that the hindbrain and motor circuitry remain intact. Instead, the effects of microglia depletion on looming-evoked escape behavior must occur further upstream in the retina, the optic tectum, or in the projections from the tectal neurons to the hindbrain. As microglia depletion induces exuberant RGC axonal arborization, one explanation for why microglia depletion disrupts defensive behavior to dark looming stimuli is that the dark looming-sensitive RGC axons cannot effectively wire with their tectal neuron counterparts, resulting in a misclassification of threatening visual stimuli when axonal pruning is disrupted. When microglia are depleted, axons form more errant connections, and dark looming-sensitive RGC axons may instead wire together with tectal neurons which are not associated with threat classification, reducing the probability that dark looming stimuli elicit escape responses. We predict that the reverse is true for bright looming-sensitive RGC axons, whereby microglia depletion results in increased likelihood of errant wiring between bright looming RGC axons and tectal neurons that classify threatening visual stimuli, leading to enhancement of escape behavior to bright looming stimuli. Conversely, it is possible that dark and bright looming-sensitive RGC axons are differentially affected by microglial depletion or the circuitry of the retina upstream of RGCs is remodeled following microglial depletion, both of which could produce the abnormal behavioral outcomes we observed. Indeed, the mechanism linking microglial depletion to improper response to looming stimuli is unclear and warrants future investigation.

## Synaptic pruning and trogocytosis are two sides of the same coin: axonal pruning

Classically, synaptic pruning has been described as the engulfment and elimination of synapses, a phenomenon dependent on the complement pathway (*Paolicelli et al., 2011*; *Perry and O'Connor, 2008*; *Schafer et al., 2012*; *Stevens et al., 2007*). While no deficit in microglial trogocytosis has been observed in ex vivo cultures from CR3 KO mice (*Weinhard et al., 2018*), one interpretation for this result is that synaptic pruning and trogocytosis are unique phenomena mediated by different mechanisms. However, our study suggests that complement-mediated synaptic pruning and trogocytosis are mechanistically related. We find that the complement system influences both axonal trogocytosis and axonal arborization, and it is possible that in CR3 KO mice, compensation through one of the other receptors for activated C3 occurs. For example, microglia express complement receptor type 4 (CR4) (*Allen Institute for Brain Science, 2015*; *Hodge et al., 2019*) which can bind

activated complement C3 to induce phagocytosis (*Ross et al., 1992*). Our data supports the hypothesis that neurons control microglial-mediated circuit remodeling through the expression of endogenous membrane-bound complement inhibitory molecules to regulate microglial trogocytosis.

# Materials and methods

## Key resources table

| Reagent type (species) or resource | Designation | Source or reference | Identifiers | Additional information |
|---|---|---|---|---|
| Gene (*Xenopus laevis*) | aRCA3 | NCBI | NCBI:XM_018246573.1 | |
| Gene (*Xenopus laevis*) | VAMP2 | Xenbase | Xenbase:vamp2.S | |
| Gene (*Xenopus laevis*) | C3 | Xenbase | Xenbase:c3.L | |
| Strain, strain background (*Escherichia coli*) | DH5α competent cells | Invitrogen | Cat#:18265017 | |
| Biological sample (*Xenopus laevis*) | Albino *Xenopus laevis* tadpoles | Nasco | RRID:XEP_Xla200 | |
| Recombinant DNA reagent | pEGFP-N1 | Clontech | Cat#: 6085–1 | |
| Recombinant DNA reagent | pFA6a pH-tdGFP | PMID:27324986 | RRID:Addgene_74322 | |
| Recombinant DNA reagent | pEF1α P2A-pHtdGFP | This paper | | Available from Edward Ruthazer upon request |
| Recombinant DNA reagent | SYP-GFP plasmid | PMID:16571768 | | |
| Recombinant DNA reagent | pEF1α SYP-pHtdGFP | This paper | | Available from Edward Ruthazer upon request |
| Recombinant DNA reagent | pEF1α aRCA3-Myc-P2A-pHtdGFP | This paper | | Available from Edward Ruthazer upon request |
| Recombinant DNA reagent | pEF1α aRCA3-mCherry-Myc-P2A-SYP-pHtdGFP | This paper | | Available from Edward Ruthazer upon request |
| Recombinant DNA reagent | pEF1α pHtdGFP-P2A-VAMP2 | This paper | | Available from Edward Ruthazer upon request |
| Recombinant DNA reagent | pEF1α pHtdGFP-P2A-Myc-VAMP2-C3 | This paper | | Available from Edward Ruthazer upon request |
| Sequenced-based reagent | RNAscope Probe against *Xenopus laevis* polr2a.L | Advanced Cell Diagnostics | RNAscope probe: Xl-polr2aL | |
| Sequenced-based reagent | RNAscope Probe against *Xenopus laevis* aRCA3 | Advanced Cell Diagnostics | RNAscope probe: Xl-LOC108708165 | |
| Commercial assay or kit | RNAscope Multiplex Fluorescent Reagent Kit v2 | Advanced Cell Diagnostics | Cat#:323136 | |
| Chemical compound, drug | PLX5622 | Plexxikon | N/A | |
| Chemical compound, drug | Polyethylene glycol 400 | Sigma | SKU:P3265 | |
| Chemical compound, drug | Poloxamer 407 | Sigma | SKU:16758 | |
| Chemical compound, drug | D-α-Tocopherol polyethylene glycol 1000 succinate | Sigma | SKU:57668 | |

*Continued on next page*

*Continued*

| Reagent type (species) or resource | Designation | Source or reference | Identifiers | Additional information |
|---|---|---|---|---|
| Chemical compound, drug | Isolectin GS-IB4 From *Griffonia simplicifolia*, Alexa Fluor 594 Conjugate | Thermo Fisher Scientific | Cat#:I21413 | |
| Chemical compound, drug | CellTracker Green BODIPY | Thermo Fisher Scientific | Cat#:C2102 | |
| Software, algorithm | STRING v11.0 protein-protein association networks | PMID:30476243 | RRID:SCR_005223 | |
| Software, algorithm | SMART protein domain annotation resource 8.0 | PMID:29040681 | RRID:SCR_005026 | |
| Software, algorithm | Phobius transmembrane topology tool | PMID:17483518 | RRID:SCR_015643 | |
| Software, algorithm | PyMOL 2.4 | Schrödinger, LLC. | RRID:SCR_000305 | *Schrödinger, LLC, 2020* |
| Software, algorithm | trRosetta | PMID:31896580 | | |
| Software, algorithm | ProBLM | PMID:25126110 | | |
| Software, algorithm | NetPhos 3.1 | PMID:15174133 | RRID:SCR_017975 | |
| Software, algorithm | R 4.0.0 | R Core Team | RRID:SCR_001905 | |
| Software, algorithm | RColorBrewer | Erich Neuwirth | RRID:SCR_016697 | |
| Software, algorithm | ComplexHeatmap | PMID:27207943 | RRID:SCR_017270 | |
| Software, algorithm | Bioconductor 3.11 | PMID:25633503 | RRID:SCR_006442 | |
| Software, algorithm | Serial Cloner 2.6.1 | SerialBasics | RRID:SCR_014513 | |
| Software, algorithm | Imaris 6 | Oxford Instruments | RRID:SCR_007370 | |
| Software, algorithm | Fluoview 5.0 | Olympus | RRID:SCR_014215 | |
| Software, algorithm | ThorImage LS | Thorlabs | | |
| Software, algorithm | Leica LAS X | Leica | RRID:SCR_013673 | |
| Software, algorithm | FIJI ImageJ | PMID:22743772 | RRID:SCR_002285 | *Schindelin et al., 2012* |
| Software, algorithm | 3D Objects Counter plugin | PMID:17210054 | RRID:SCR_017066 | |
| Software, algorithm | 3D ROI Manager plugin | PMID:23681123 | RRID:SCR_017065 | |
| Software, algorithm | TrackMate plugin | PMID:27713081 | | |
| Software, algorithm | Descriptor-based series registration plugin | PMID:20508634 | | |
| Software, algorithm | 3D hybrid median filter plugin | Christopher Philip Mauer and Vytas Bindokas | | |
| Software, algorithm | ScatterJ plugin | PMID:25515182 | | |
| Software, algorithm | MATLAB | MathWorks | RRID:SCR_001622 | |
| Software, algorithm | CANDLE | PMID:22341767 | | |
| Software, algorithm | Graphpad Prism 9 | GraphPad Software | RRID:SCR_002798 | |
| Software, algorithm | Fisher r-to-z transformation calculator | VassarStats | RRID:SCR_010263 | |
| Software, algorithm | Cura 4 | Ultimaker | RRID:SCR_018898 | |
| Software, algorithm | TinkerCAD | Autodesk | | |
| Software, algorithm | Psychopy 3.0 | PMID:30734206 | RRID:SCR_006571 | |

*Continued on next page*

*Continued*

| Reagent type (species) or resource | Designation | Source or reference | Identifiers | Additional information |
|---|---|---|---|---|
| Software, algorithm | XenLoom (beta): Looming Stimulus Presentation and Tracking of *Xenopus laevis* tadpoles | This paper | | github.com/ tonykylim/ XenLoom_beta; *Lim, 2021*; copy archived at swh:1:rev:b487791d 1d91a5950eeb fd1e7640e 0c3db761cf5 |
| Other | Allen Brain Map Human Multiple Cortical Areas SMART-seq data set | Allen Institute for Brain Science | | |
| Other | Estink 2000 lumens mini LED projector | Amazon.ca | ASIN#:B07F7RT9XZ | |
| Other | 96 white 20 lb bond copy paper | Staples.ca | Item#:380480 | |
| Other | Custom 3D printed mount for projector lens | This paper | | http://www.thingiverse. com/thing:4335379 |
| Other | Custom 3D printed stage for *Xenopus laevis* behavior | This paper | | http://www.thingiverse. com/thing:4335395 |
| Other | 8-inch soda lime 1500 ml culture dish | Carolina | Item#:741006 | |
| Other | 60 mm petri dish | Fisher Scientific Canada | Cat#:FB0875713A | |
| Other | PLA 1.75 mm 3D printing filament | iPrint-3D | Transparent purple | |
| Other | Logitech C920 webcam | Logitech | Model#:960–000764 | |
| Other | Anycubic i3 mega 3D printer | Amazon.ca | ASIN#:B07NY5T1LJ | |

## Lead contact and materials availability

Plasmids generated in this study will be made available upon request. Further information and requests for resources and reagents should be directed to the Lead Contact, Edward Ruthazer (edward.ruthazer@mcgill.ca).

## *Xenopus laevis* tadpoles

Adult albino *Xenopus laevis* frogs (RRID:XEP_Xla200) were maintained and bred at 18°C. Female frogs were primed by injection of 50 IU pregnant mare serum gonadotropin (ProSpec-Tany Techno-Gene Ltd., Ness-Ziona, Isreal). After 3 days, male and primed female frogs were injected with 150 IU and 400 IU of human chorionic gonadotropin (Sigma-Aldrich, Oakville, CA) into the dorsal lymph sac, respectively. The injected male and female frogs were placed in isolated tanks for mating. Embryos were collected the following day and maintained in Modified Barth's Saline with HEPES (MBSH) in an incubator at 20°C with LED illumination set to a 12 hr/12 hr day-night cycle and staged according to Nieuwkoop and Faber (NF) developmental stages (*Nieuwkoop and Faber, 1994*). All experiments were conducted according to protocol application number 2015–7728 approved by The Animal Care Committee of the Montreal Neurological Institute and in accordance with Canadian Council on Animal Care guidelines. *Xenopus laevis* sex cannot be determined visually pre-metamorphosis, and thus the sex of experimental animals was unknown.

## Labeling of microglia, axons, and brain structures

Stage 39/40 tadpoles were anesthetized with 0.02% MS-222 in 0.1X MBSH. For labeling of microglia, tadpoles received an intracerebroventricular (icv) injection to the third ventricle of 1 mg/ml Alexa 594 conjugated IB4-isolectin. A minimum of 48 hr was allowed to pass before live imaging studies were commenced to allow for binding and update of IB4-isolectin by microglial cells. For concurrent labeling of brain structures, 1 mM CellTracker Green BODIPY in 10% DMSO was injected icv after

isolectin injection. For labeling of RGC axons, electroporation of plasmid DNA into RGC cells was performed as described previously (*Ruthazer et al., 2006*; *Ruthazer et al., 2013a*). In brief, stage 39/40 tadpoles were anesthetized with 0.02% MS-222 in 0.1X MBSH. A glass micropipette was back filled with endotoxin free maxi-prep plasmid solution (2–3 µg/µl) and fast green to visualize the injection. The micropipette was advanced into the eye, and DNA solution pressure injected. The micropipette was then withdrawn, and parallel platinum electrodes were placed to bracket the eye. For bulk labeling of RGCs, a Grass Instruments SD9 electrical stimulator was used to apply 4–6 pulses of 2.4 ms duration at 36 V with a 3 µF capacitor was placed in parallel to obtain exponential decay current pulses. For single-cell labeling of RGCs, 4–6 pulses of 1.6 ms duration at 36 V was applied.

### Microglial depletion

PLX5622 was dissolved in DMSO at 20 mM, aliquoted, and stored at −20°C. Thawed aliquots of PLX5622 were sonicated briefly for 1 min before dilution in polyethylene glycol (PEG) 400. PEG 400 solution was then further diluted in a solution of 0.1X MBSH containing non-ionic surfactants poloxamer 407 and D-α-tocopherol polyethylene glycol 1000 succinate (TPGS) to form a mixed micelle drug delivery vehicle (*Guo et al., 2013*). Final vehicle rearing solution consisted of 2.5% PEG 400, 0.04% poloxamer 407, 0.01% TPGS, and 0.05% DMSO in 0.1X MBSH. Initially, animals were reared in 0.1X MBSH. At stage 35–40, animals were transferred to vehicle rearing solution with or without 10 µM PLX5622 for rearing. Rearing solutions were refreshed daily with newly prepared drug and vehicle solutions.

### Two-photon live imaging

Two-photon live imaging of axons and microglia cells was performed as described previously (*Ruthazer et al., 2013b*). Tadpoles were placed in a Sylgard 184 silicone imaging chamber with their dorsal side facing up and were covered with a #1 thickness glass coverslip. One µm interval z-stacks of the optic tectum were collected on a custom-built two-photon microscope (Olympus BX61WI with Olympus FV300 confocal scan head) outfitted with an Olympus 1.0 NA 60x water-immersion objective, or a Thorlabs multiphoton resonant scanner imaging system outfitted with an Olympus 1.0 NA 20x water-immersion objective. Excitation was produced using a Spectra-Physics InSight3X femtosecond pulsed laser. Images were collected at 512 × 512 pixels with Fluoview 5.0 or ThorImage LS.

For real-time imaging studies, stage 46–48 tadpoles were immobilized by brief (2–8 min) immersion in freshly thawed 2 mM pancuronium bromide in 0.1X MBSH. Animals were then maintained in 0.1X MBSH for imaging. Z-stacks were collected at 6 min intervals. For daily imaging studies, stage 43–47 tadpoles were anesthetized by immersion in 0.02% MS-222. Z-stacks were collected, and animals were returned to 0.1X MBSH rearing solution.

When imaging microglia (Alexa 594 conjugated IB4-isolectin) and RGC axons (eGFP or pHtdGFP) concurrently, excitation wavelength was set at 830 nm and a 565 nm emission dichroic was used in conjunction with green (500–550 nm) and red (584–676 nm) filters for fluorescence emission detection on separate photomultiplier tubes. When imaging microglia (Alexa 594 conjugated IB4-isolectin) and brain structures (CellTracker Green BODIPY), imaging was done sequentially. Alexa 594 was first imaged with excitation wavelength set to 810 nm and fluorescence emission detection through the red filter. Imaging of CellTracker Green BODIPY was then performed at excitation wavelength 710 nm and the fluorescence emission detected through the green filter. When imaging RGC axons alone (pHtdGFP), excitation wavelength was set to 910 nm and fluorescence emission detected through the green filter. When imaging aRCA3-mCherry-Myc and SYP-pHtdGFP, images were captured concurrently at wavelength 990 nm through the red and green filters, respectively.

### Two-photon laser-induced injury

Laser irradiation injury was carried out by repeatedly scanning an approximately 10 × 10 µm region under high laser power at wavelength 710 nm for 10 s.

### Microscopy image processing and analysis

For visualization purposes, the 3D object counter (*Bolte and Cordelières, 2006*) and 3D ROI manager (*Ollion et al., 2013*) plugins were used on z-stacks for segmentation and effacement of

melanophores on the dorsal dermal surface of the animal before maximum intensity projection. Analysis was performed by an experimenter blind to treatment group.

## Axon morphology analysis

Two-photon z-stacks were captured daily from 2 to 5 days post-electroporation. Z-stacks were denoised with CANDLE (*Coupé et al., 2012*), and were manually traced using Imaris six software.

## Microglia quantification

The number of microglia and microglia process numbers were counted manually within 150-µm-thick z-stacks of the optic tectum.

## Microglia mobility

For microglial mobility experiments, microglia in the optic tectum were imaged over 2 hr at 6 min intervals and manual tracking of microglia was carried out using the TrackMate plugin (*Tinevez et al., 2017*).

## Real-time imaging of trogocytosis of axons by individual microglial cells

Collected z-stacks were denoised with the 3D hybrid median filter plugin. Time series were registered by descriptor based series registration (*Preibisch et al., 2010*). Microglia 3D region of interests (ROIs) were generated using the 3D object counter plugin (*Bolte and Cordelières, 2006*) using the same threshold across all time points. When microglia 3D ROIs were overlapping, they were split manually. Green fluorescence intensity within microglia 3D ROIs was measured using the 3D ROI manager plugin (*Ollion et al., 2013*) and background subtraction was carried out with a rotated or mirrored z-stack. Baseline values were defined as the average of the first 15 frames, except in the case of microglial interaction, wherein any frames after an interaction were excluded from baseline calculation. Movies were generated using Imaris software or ImageJ.

## Trogocytosis assay

Microglia were labeled with Alexa 594-conjugated isolectin and axons with pHtdGFP or SYP-pHtdGFP as described above. On day 4 and day 5 post-labeling, 150-µM-thick z-stacks were collected from the optic tectum. Laser power and photomultiplier tube voltage was kept constant on both days. The number of labeled axons was counted manually. In studies when axons were labeled with pHtdGFP, data were excluded if axonal blebbing was detected, or if the number of labeled axons fell from day 4 to day 5. This data exclusion step was not performed when axons were labeled with SYP-pHtdGFP. Using the red microglia channel, microglia 3D ROIs were automatically generated using the 3D object counter plugin (*Bolte and Cordelières, 2006*). Microglia segmentation threshold was set to the mean pixel intensity of the red channel plus two standard deviations. A minimum threshold size of 1500 pixels was used to exclude background from analysis. The 3D ROI manager plugin (*Ollion et al., 2013*) was then used to measure the mean gray value of the green channel within microglial cell ROIs. To reduce background signal in the green channel, the mode of the green pixel intensity was calculated from the z-stack and subtracted. Non-microglial ROIs such as melanophores as well as microglia which had overlapping pixels with the axon were excluded manually and the average microglial green fluorescence intensity for the population of microglia in the z-stack was determined.

## Colocalization analysis

Two-photon z-stacks were captured on day three post-electroporation. aRCA3-mCherry-Myc and SYP-pHtdGFP channels were denoised using the 3D hybrid median filter ImageJ plugin. A 3D mask of the axon was generated by summing the red and green channels and using an intensity threshold for segmentation of the axon. Pearson's correlation coefficient (*Manders et al., 1993*) of the masked axon was calculated using Imaris software, with thresholds determined automatically using the method of Costes (*Costes et al., 2004*). The ScatterJ plugin (*Zeitvogel et al., 2016*) was used to generate scatter plots.

## Looming stimulus behavioral assay

### Experimental setup

Stage 47 tadpoles were placed within a closed 60 mm petri dish and allowed to swim freely. The petri dish was placed on the bottom of a large shallow (20 cm diameter, 8 cm deep) glass culture dish filled completely with 0.1X MBSH. The large shallow glass culture dish was placed on a purple 3D printed stage to allow for automated segmentation of albino tadpoles. A webcam was placed above the tadpole to record tadpole behavior, while a 2000 lumens projector customized by 3D printing to shorten the focal distance, was used to project visual stimuli onto a piece of 96 white 20 lb bond copy paper taped to the side of the large glass culture dish.

### Stimulus presentation and recording

Dark or bright looming stimuli were presented using custom code written in Python 3. An expanding circle was projected onto the culture dish (800 × 600 pixels or 10.6 × 8 cm) using the PsychoPy library (*Peirce et al., 2019*). For dark looming stimuli, a black (29 cd/m$^2$) expanding circle was shown over a white (208 cd/m$^2$) background. For bright looming stimuli, a white expanding circle was shown over a black background. During looming, the diameter of the circle expanded exponentially at 10% per frame at 60 frames per second, from a diameter of 54 pixels (7 mm) until it encompassed the entire screen 0.5 s later. After another 0.8 s, the screen was reset and a 10 s refractory period commenced. In parallel to stimulus presentation, 480 p video was recorded by webcam using the OpenCV library (*Bradski, 2000*). Ten looming stimulus trials per animal were recorded.

### Automated tracking of tadpole behavior

Custom computer vision tadpole tracking code was written using Python three and OpenCV (*Bradski, 2000*). Feature detection on the petri dish was used to automatically determine the scale of video data. Background was subtracted and segmentation of the tadpole was carried out by thresholding. The resulting mask was fit to an ellipse to extract location and speed, as well as directional data and escape angle. Location data from 0 to 2 s following the onset of the looming stimulus was summarized and displayed as a contrail. Instantaneous velocity for the 3 s before and after the looming stimulus was also extracted and compared by area under curve analysis to obtain distance traveled.

### Discrimination of positive and negative responses to looming stimuli

An automated python script was used to randomize tadpole videos and a user blinded to treatment categorized tadpole responses to looming stimuli as defensive escape behavior (positive response), absence of defensive escape behavior (negative response), or undeterminable (excluded from response rate calculation). Tadpole responses were categorized as undeterminable if the tadpole was moving quickly when the looming stimulus was presented. The response rate was calculated as the number of positive responses divided by the combined number of positive and negative responses.

### Assessment of motor responses

Positive escape responses were further analyzed to compare motor data. Escape distance and maximum velocity over 2 s from the onset of the loom was calculated. Escape angle was also measured from heading data. In dark looming stimuli trials, the escape angle was defined as the absolute value of the change in heading from 0 to 0.6 s. For bright looming stimuli trials, the escape angle was defined as the absolute value of the change in heading from 0 to 1.2 s. This difference is because dark looming stimuli evoked escape movement beginning approximately 0.5 s after looming onset, whereas bright looming stimuli evoked escape movement beginning approximately 1.0 s after looming onset.

## Bioinformatics analysis

### Identification of complement inhibitory proteins

To identify candidate complement inhibitory proteins, the STRING database (*Szklarczyk et al., 2019*) was used to look for functional interactions between complement C3 and other proteins.

Human complement C3 was queried using a medium confidence (0.4) minimum interaction score, and a limit of 10 and 20 interactors for the first and second shell, respectively. Sources was limited to textmining, experiments and databases.

### Expression of complement inhibitory proteins in human cortical cells

The Allen Brain Map Human Transcriptomics Cell Types Database (*Allen Institute for Brain Science, 2015*) was used to examine the expression of candidate proteins identified from the STRING query. Cell taxonomy and hierarchical clustering was retained from previous analysis (*Hodge et al., 2019*). Gene expression of complement inhibitory proteins across distinct brain cell clusters was examined and plotted using R software (*R Development Core Team, 2018*) and the Bioconductor (*Huber et al., 2015*), ComplexHeatmap (*Gu et al., 2016*), and ColorBrewer (*Neuwirth, 2014*) packages.

### Homology search

The protein sequence of human CD46 (NCBI accession #: NP_002380.3) was queried in the *Xenopus laevis* genome with NCBI protein-protein BLAST (*Altschul et al., 1990*). The top three hits were predicted proteins XP_018102062.1, XP_018100169.1 and XP_018102058.1. The identity of the predicted proteins was obtained by cross-referencing with the *Xenopus tropicalis* genome using BLAST.

### Conserved modular architecture analysis and transmembrane topology

The SMART protein domain research tool (*Letunic and Bork, 2018*; *Schultz et al., 1998*) was used to predict conserved modular architecture and the Phobius transmembrane topology tool (*Käll et al., 2004*; *Käll et al., 2007*) was used to predict transmembrane topology.

### Tertiary protein structure prediction

De novo tertiary protein structure was predicted using trRosetta (*Yang et al., 2020*). The extracellular region, transmembrane region, and intracellular regions of human CD46 and *Xenopus laevis* aRCA3 were modeled separately by trRosetta and joined using pyMOL (*Schrödinger, LLC, 2020*) The orientation of the protein in the phospholipid bilayer was predicted using the protein bilayer lipid membrane orientation package (*Kimmett et al., 2014*).

### Kinase prediction

The NetPhos 3.1 tool (*Blom et al., 2004*) was used to make predictions of tyrosine phosphorylation sites on the intracellular C-terminus of aRCA3. A cutoff score of 0.6 was used.

## RNAscope in situ hybridization

### Tissue preparation

The RNAscope (*Wang et al., 2012*) fixed-frozen tissue sample preparation and pretreatment protocol provided by the manufacturer was modified for *Xenopus laevis* tadpoles. Stage 46 tadpoles were euthanized in 0.2% MS-222 in 0.1X MBSH. Tadpoles were then fixed in 4% PFA at 4°C for 24 hr on a laboratory rocker. For cryoprotection, tadpoles were then moved sequentially through 10%, 20% and 30% sucrose in 1X PBS, until samples sunk to the bottom of the container. Cryoprotected tadpoles were then embedded in OCT blocks on dry ice. OCT blocks were sectioned at 8 μm thickness on a cryostat and mounted on superfrost plus slides. To enhance tissue adhesion, slides were air dried for 2 hr at −20°C and baked at 60°C for 30 min. Slides were then post-fixed by immersion in 4% PFA for 15 min, and then dehydrated with 50%, 70% and 100% ethanol. Slides were treated with hydrogen peroxide to block endogenous peroxidases. For target retrieval, using a hot plate and beaker, slides were boiled in RNAscope target retrieval reagent and then treated with RNAscope Protease III.

### RNAscope assay

The RNAscope Multiplex Fluorescent 2.0 Assay was performed according to manufacturer's protocols using the HybEZ oven. In brief, probes were applied to slides, and three amplification steps were carried out. Opal 570 dye was applied to slides along with Hoechst counterstain. Coverslips

were mounted with Prolong gold antifade mountant. Slides were imaged with a Leica TCS SP8 confocal.

## Molecular biology

The following plasmid constructs were constructed and used in this study: pEF1α-pHtdGFP; pEF1α-SYP-pHtdGFP; pEF1α-aRCA3-Myc-P2A-pHtdGFP; pEF1α-aRCA3-mCherry-Myc-P2A-SYP-pHtdGFP; pEF1α-pHtdGFP-P2A-Myc-VAMP2-C3; pEF1α-pHtdGFP-P2A-VAMP2.

### cDNA library

Stage 40 Tadpoles were euthanized in 0.2% MS-222 in 0.1X MBSH. Animals were transferred to TRIzol reagent (Invitrogen, Carlsbad, CA) and homogenized by sonication and processed according to manufacturer's protocols to isolate mRNA. Superscript IV reverse transcriptase (Invitrogen) was then used according to manufacturer's protocol to generate whole tadpole cDNA.

### Cloning and plasmid isolation

Cloning of *Xenopus laevis* cDNA was carried out using primers flanking genes of interest. Primers were generated according to mRNA sequences for aRCA3 (XM_018246573.1), VAMP2.S (NM_001087474.1), and c3.L (XM_018253729.1) predicted by NCBI (*NCBI Resource Coordinators, 2018*) and Xenbase (*Karimi et al., 2018*). Synaptophysin was subcloned from mouse synaptophysin (*Ruthazer et al., 2006*). pHtdGFP was subcloned from a pFA6a plasmid (*Roberts et al., 2016*). PCR was performed with Phusion High-Fidelity DNA polymerase (Thermo Scientific, Waltham, MA) and DNA fragments were cut with NEB enzymes and ligated into a plasmid with an EF1α promotor, ampicillin resistance, and a P2A self-cleaving construct or Myc tag when appropriate. DH5α bacteria were transformed and single clone colonies containing inserts were isolated. Endotoxin-free plasmid preparations of high yield and purity for electroporation was prepared by maxi-prep (Qiagen, Hilden, Germany).

### Design of fusion proteins

The SYP-pHtdGFP fusion protein was created by in-frame fusion of pHtdGFP to the C-terminus of SYN between a three amino acid linker (QGT). The aRCA3-mCherry-Myc fusion protein was created by an in-frame fusion of mCherry-Myc to the C-terminus of aRCA3 between a two amino acid linker (AC). The Myc-VAMP2-C3 fusion protein was created by an in-frame fusion of Myc to the N-terminus of VAMP2 between a four amino acid linker (PGKI) and an in-frame fusion of C3 to the C-terminus of VAMP2 between an 18 amino acid linker (ASIKSPVQPLSAHSPVCI). The long linker for VAMP2-C3 was chosen to ensure that the C-terminus transmembrane domain of VAMP2 would not sterically hinder proper folding of C3.

## Statistical analysis

Statistical analysis was performed using Graphpad Prism nine software. All data is presented as mean ± SEM, unless otherwise noted. Data was assessed for normality by D'Agostino-Pearson omnibus test and examination of quantile-quantile plots and assessed for homoscedasticity by examination of residual plots. All statistical tests were two-sided, and differences were considered significant for $p < 0.05$.

### Pairwise comparisons

When normality and homoscedasticity assumptions were met, differences between two groups were tested by unpaired t-test. Welch's t-test was used if the assumption of homogeneity of variance was not met. Non-normally distributed pairwise comparisons were tested by Mann-Whitney test.

### Group data

If no datapoints were missing, two-way repeated measures ANOVA was used to test group data. If an interaction was found, the interaction was reported and Sidak's multiple comparison test was used to test between groups or time points as appropriate. If no interaction was found, then the main effect was reported. Conversely, if the data contained missing values (due to loss of animals during a time course), the mixed effect restricted maximum likelihood model was used.

## Correlation

Correlations were calculated using Pearson's correlation coefficient. Trendlines were plotted using simple linear regression. Significance testing of the difference between two correlation coefficients was carried out using Fisher's z transformation.

## Non-linear curve fitting

Log-transformed histograms were fit by Poisson regression with no weighting. The means of the best-fit curves were compared by likelihood-ratio test.

## Data and code availability

The quantitative data used for analyses and figure generation in this paper are publicly available at https://doi.org/10.6084/m9.figshare.12895172.v1. The documented source code and user guide for the looming stimulus presentation and tadpole tracking software module, XenLoom (Beta), developed in this study are available at https://github.com/tonykylim/XenLoom_beta.

# Acknowledgements

We thank Dr Wayne Sossin, Dr Jean-Francois Cloutier and the MNI Microscopy Core Facility for sharing access to equipment. We also thank Dr Philip Kesner and Anne Schohl for technical advice, Dr Larissa Ferguson for technical assistance, and all members of the Ruthazer lab for helpful discussions. The pFA6a-pHtdGFP plasmid was generously provided by Dr Joerg Stelling. PLX5622 was kindly provided by Plexxikon, Inc TKL was supported by a CIHR Postdoctoral Fellowship and the McGill Faculty of Medicine McLaughlin Postdoctoral Fellowship. This work was supported by grants to ESR from FRQS (31036) and CIHR (FDN-143238).

# Additional information

## Funding

| Funder | Grant reference number | Author |
| --- | --- | --- |
| Canadian Institutes of Health Research | FDN-143238 | Edward S Ruthazer |
| Fonds de Recherche du Québec - Santé | 31036 | Edward S Ruthazer |
| Canadian Institutes of Health Research | Postdoctoral fellowship | Tony KY Lim |
| Faculty of Medicine, McGill | Postdoctoral Fellowship | Tony KY Lim |

The funders had no role in study design, data collection and interpretation, or the decision to submit the work for publication.

## Author contributions

Tony KY Lim, Conceptualization, Resources, Software, Formal analysis, Funding acquisition, Investigation, Visualization, Methodology, Writing - original draft; Edward S Ruthazer, Conceptualization, Resources, Data curation, Formal analysis, Supervision, Funding acquisition, Methodology, Project administration, Writing - review and editing

## Author ORCIDs

Tony KY Lim (ID) https://orcid.org/0000-0003-1843-0060
Edward S Ruthazer (ID) https://orcid.org/0000-0003-0452-3151

## Ethics

Animal experimentation: All experiments were conducted according to protocol application number 2015-7728 approved by The Animal Care Committee of the Montreal Neurological Institute and in accordance with Canadian Council on Animal Care guidelines.

Decision letter and Author response
Decision letter https://doi.org/10.7554/eLife.62167.sa1
Author response https://doi.org/10.7554/eLife.62167.sa2

## Additional files

### Supplementary files
- Supplementary file 1. Table of all plasmids used and generated.
- Transparent reporting form

### Data availability
Data used for analysis and figure generation are included in the manuscript and supporting files. Source data files have been provided for Figures 1–8.

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
