## [Decision Letter]

**Acceptance summary:**

Your study on microglial pruning of axon endings is excellent and well documented. Using the developing retinotectal projection of *Xenopus laevis* tadpoles as a model system, you have used molecular manipulations, live imaging and behavior, to provide compelling in vivo evidence that microglia prune axons via trogocytosis – a partial engulfment and breakdown of axon terminals, identifying this process as an important regulator of axonal pruning to promote correct neural wiring during development.

**Decision letter after peer review:**

Thank you for submitting your article "Microglial trogocytosis and the complement system regulate axonal pruning in vivo" for consideration by *eLife*. Your article has been reviewed by three peer reviewers, and the evaluation has been overseen by a Reviewing Editor and K VijayRaghavan as the Senior Editor. The following individuals involved in review of your submission have agreed to reveal their identity: Carlos Aizenman (Reviewer #1); Cornelius T Gross (Reviewer #2); Amanda Sierra (Reviewer #3).

The reviewers have discussed the reviews with one another and the Reviewing Editor has drafted this decision to help you prepare a revised submission.

Summary:

Your study linking microglial trogocytosis, or partial engulfment of axon endings, to the maturation of axonal terminal arborizations in the developing *Xenopus* retinotectal system was in general praised by three reviewers. The longitudinal in vivo fluorescent imaging to simultaneously monitor microglia motility and retinal ganglion cell axon terminal branch dynamics in the optic tectum are superb and informative. Examining the functional consequences of this process, as well as relating the process of trogocytosis to the complement system, are solid additions to your study.

The reviewers had some issues, nevertheless, on controls for autofluorescence and for the complement experiments, literature citation and discussion, and in some cases, overinterpretation of your data. Most of their comments can be readily amended with your existing data. Below, is a summary of their comments and suggested revisions, taken from the appended reviews and from the reviewer discussion, which was more interactive and thorough than usual with the reviewers ultimately agreeing with each other, all to your benefit.

Revisions:

1) Measuring changes in branching versus axon length: Reviewer 2 questioned whether microglia depletion results only in a change of branching, but not overall length of axons. During the reviewer discussion, reviewer 1 pointed out that the changes observed occur over a short time course, of small lengths of an otherwise large arbor and so changes in axon length would not be a useful measure. I agree. This point should be taken up in the Discussion.

2) Controls for autofluorescence, to demonstrate true trogocytosis signal vs autofluorescence:

a) As suggested by reviewer 3, is it possible to quantify with appropriate negative controls (for instance, experiments with unlabeled axons, or with off-target labeling in other brain structures) to demonstrate the specificity of the quantification?

b) Reviewer 2 suggests that traces for all individual cells in the validation experiments (Figure 2F) should be shown to understand the variability in the persistent change in GFP signal that remains after contacts are completed. In addition, a spectral analysis should be done to help rule out autofluorescence.

c) Correlative EM, also suggested, would be a labor-intensive addition, and is not deemed necessary if one of the former suggestions are taken.

3) Complement experiments

a) Reviewer 2 thought that the proper control for the VAMP2-C3 overexpression is VAMP2 itself, is not ideal, citing that VAMP2-C3 fusion might have dominant negative effects that could have little to do with C3; also, that a better control would be VAMP2-C3* where C3* is non-cleavable or even a VAMP2-GFP fusion. He also wondered why you did not use a GPI anchor if the idea was to present C3 on the membrane, and whether you had evidence that C3 in this fusion form is functional. In the Consultation session, reviewer 1 did not think that the VAMP-GFP fusion or uncleaveable C3 would be better, and surmised that you used VAMP rather than GPI anchors to limit expression to presynaptic terminals. Reviewer 2 accepts this point but please discuss the inherent weaknesses in using the VAMP2-C3 control, either in the Materials and methods and/or Results.

b) Reviewer 3 thought that for the aRCA3 overexpression and VAMP2-C3 expression experiments (Figures 7 and 8), you should show direct evidence that trogocytosis is altered by quantifying the axonal fluorescent signal in microglia (as in Figure 3C), as proof that engulfment is potentiated or decreased (instead of showing only the downstream effect in axons). Reviewer 1 rebutted this comment by indicating that he did see a difference in axon phenotypes for depletion and CD46 homolog or C3 overexpression. He also acknowledges that you do discuss and invoke compensations due to the cell autonomous nature of the CD46 homolog and C3 over-expression. However, he did find it disappointing that you did not follow up on a single axon level with longitudinal in vivo live imaging what the impact of microglia contacts is in these circumstances. Please address the shortcomings of these experiments in the Discussion.

c) Reviewer 2, # 6 comment on why depletion of microglia does not alter the number of axons, only their branching, but manipulations of complement seem to alter total numbers of axons. Please cite the possibility that C3 has an impact on microglia-mediated neuron elimination via apoptosis (see on RGC apoptosis and C3r, Anderson et al., 2019). Reviewer 1 thought that both microglia depletion and complement manipulation altered number of axonal branches, not number of axons. Please address this point in the Discussion.

4) RGC subtype specificity; pruning in areas outside the tectum:

a) As per reviewer 1's comments, that because there is a bigger effect on dark looming detectors than light looming detectors, different types of axons (from different RGCs having different response selectivity) could be differently selective to microglial pruning.

b) Inhibition of microglial activation by PLX5622 might also affect remodeling in visual circuits, and could account for the behavioral difference.

Please discuss.

5) Unease with the behavioral results: Reviewer 2, comment #4 and Figure 9: the "behavioral part of the manuscript is only weakly linked to the main story". Reviewer 1 somewhat concurs, especially since this behavior is something that is known to require intact tectal circuitry. To this end, reviewer 3 proposes that to ensure that microglia outside the optic tectum, including the retina, regions involved in locomotion, as well as peripheral macrophages are not affected, evidence should be provided that the depleted animals do not have impairments in their vision, locomotor activity, and overall health. Reviewer 2 subsequently agreed to include the behavior, provided that you explicitly state that the mechanism that links depletion to behavior is far from understood, and remove Figure 9 summary, as it now stands, "to offset the abundance of correlative findings and hype in this field and reinforcement of (such) data". Reviewer 3 adds that you could modify Figure 9 to summarize what is known and what is not, and put question marks whenever appropriate. The latter would be a compromise position.

6) Literature citation: All three reviewers ask for a more critical appraisal of the literature regarding microglial synaptic pruning, to highlight what you have done so well and how it is novel (see Reviewer 2, point #9; Reviewer 3: Point #1): Although previous and widely cited studies argue for microglial pruning, based on KO and depletion models, there is little direct evidence that microglia actively select the synapses to be removed (in contrast to neural shdding of synapses, later to be engulfed by microglia). Other live imaging studies have failed to show that microglia grasp and digest synapses from living neurons; this includes the Weinhard and Gross study using organotypic cultures. They instead report axonal trogocytosis, for the first time, in vivo. Yours is the first study very nicely demonstrating trogocytosis by live imaging.

Reviewer #1:

This is a beautiful study, and probably one of the best work I've seen from this group, already known for excellent, high-quality papers in developmental neuroscience.

The authors use an in vivo system to examine the hypothesis, based on prior in vitro data, that microglia prune axons via troglocytosis – a partial engulfment of an axon, basically by engulfing and breaking down axon terminals. The advantages of this in vivo system is that they can also push this hypothesis much further by examining the functional implications of microglial troglocytosis, and directly link the role of the complement system in synaptic pruning to microglial engulfment of axon terminals.

Using the developing retinotectal projection of *Xenopus laevis* tadpoles as a model system, the authors present a compelling set of experiments, that include molecular manipulations, live imaging and behavior, to provide compelling in vivo evidence that microglia troglocytose presynaptic axon terminals, and that this is an important regulator of axonal pruning to promote correct neural wiring during development.

Their experiments are as follows: *Xenopus* microglia are identified by labelling with IB4-isolectin, and using in vivo imaging are found to me highly motile and responsive to injury, consistent with known MG functions. Remarkably MG enter the neuropil and are shown to associate with a green FP labelled retinal axon and depart with little bits of green fluorescence in their bellies. Moreover, using labelled synaptophysin (a synaptic vesicle protein) they show that MG are actually engulfing synaptic terminals. This increase in green fluorescent in the MG was shown to be highly significant as shown as an increase over a day. These experiments convincingly show troglocytosis by MG of synaptic terminals in vivo and in real time. The provided videos are incredible.

Next they look at the developmental consequences of MG pruning. In the first experiment they deplete MG using an inhibitor of a kinase necessary for MG survival. Depletion of MG results in increased branching of RGC axon terminals, suggesting that MG are negatively regulating axon branching over development. To test the consequences of this, they use a behavioral response to either dimming or brightening visual stimuli. Remarkably, MG depletion induced remodeling suppresses responses to dimming stimuli and enhances responses to brightening stimuli. While this is a puzzling finding, it does show that disrupted remodeling does cause abnormal behavior.

To find whether the complement system is important for regulating MG pruning, they first look for complement regulating proteins known to be associated in the brain, and identify human CD46 as a top candidate. The closes protein to this was the frog aRCA3 protein, a transmembrane protein with several complement control protein domains (CCCPs). Using RNAscope in situ hybridization, they find that aRCA3 is expressed in the RGC layer in the retina, and that it co-localizes with RGC axon terminals. By overexpressing aRCA2 (and thus inhibiting complement) they see decreased MG troglocytosis and enhanced axonal branching, whereas increasing complement expression in axons (by anchoring it to the presynaptic terminal) they saw increased pruning.

Together these data show compelling evidence that the complement system regulates activation of microglia, and thus axonal pruning, which results in normal sculpting of axonal projections in the developing brain.

I really struggled to find something I didn't like about this study. The data is high-quality, the logic is clear, the findings are incredibly cool and compelling. This paper will be of interest to a wide range of scientists, including developmental neuroscientists, systems neuroscientists, those interested in the role of the immune system in the brain, etc. What I especially like is how it brings together a bunch of disparate ideas and creates a sense of clarity that makes it one of those papers that read as an instant classic.

Reviewer #2:

This manuscript by Lim et al. takes important steps to link microglia to the maturation of axonal terminal arborization in the *Xenopus* vertebrate model. They take advantage of the unique power of this system to perform longitudinal in vivo fluorescent imaging to simultaneously monitor microglia motility and retinal ganglion cell axon terminal extensions in the optic tectum. Their major novel observation is that microglia appear to persistently retain cytoplasmic GFP signal expressed in axons following their contacts with axons. They invoke microglia-neuron trogocytosis as a possible mechanism to explain this phenomenon. They then go on to show that pharmacological depletion of microglia results in an increase in axonal branching and conjecture that trogocytosis is involved in reducing axonal branching by a trogocytosis-dependent pruning phenomenon. In a digression they also show that the same pharmacological treatment results in changes in behavioral responses to dark and light looming stimuli. In the second half of the work they overexpress a *Xenopus* CD46 homolog and show that it reduces GFP retention and increases axonal outgrowth/branching and overexpress C3 and show that it decreases GFP retention and decreases axonal outgrowth/branching. The authors conclude that microglia restrict axon terminal outgrowth by trogocytosing axon material via a complement-dependent mechanism.

The work makes several important observations and has the potential to link microglia function to the pruning of axon terminals. In particular, several observations are important for the field in my opinion and could be elaborated more, including the finding that microglia depletion results in a change of branching, but not overall length of axons, and the report of frequent switching back and forth between ramified and amoeboid morphologies and entry and exit from the neuropil layer. However, several data are overinterpreted in my view and the authors need to hone closer to the findings so as to emphasize their most important contributions.

Issues that should to be addressed:

1) The GFP retention assay needs controls in order to rule out confounds due to the background autofluorescence mentioned by the authors. Traces for all individual cells in the validation experiments (Figure 2F) should be shown for the reader to understand the variability in the persistent change in GFP signal that remains after contacts are completed. In addition, a spectral analysis should be done to help rule out autofluorescence.

2) To causally link GFP retention to axonal remodeling it would be helpful to follow the axonal arborization of neurons that have been contact by microglia. Do they change in a manner consistent with the microglia depletion experiments (changes in branching but not axon length)? Without this kind of data it is hard to understand the impact of microglia contacts on axons.

3) Figure 3EF: it seems odd that Syp-GFP is taken up by microglia as it should be primarily localized to vesicles. Does this suggest that microglia engulf axons with their vesicles inside?

4) The behavioral part of the manuscript is only weakly linked to the main story. Microglia depletion will clearly have multiple effects on development beyond the remodeling of axons in the optic tectum and without a more restricted depletion or manipulation of microglia and documented interpretable changes in the responsible circuitry, the link to changed behavior is tenuous. Given this weak mechanistic link, Figure 9 is premature and misleading and should be eliminated.

5) It is not clear that the proper control for the VAMP2-C3 overexpression is VAMP2 itself. I'd be concerned that a VAMP2-C3 fusion might have dominant negative effects that could have nothing to do with C3 and could explain the difference with VAMP2. The better control would be VAMP2-C3* where C3* is non-cleavable or even a VAMP2-GFP fusion. Also, if the idea was to present C3 on the membrane then why didn't the authors use a GPI anchor? Moreover, it would be good to have evidence that the C3 in this fusion form is in fact functional.

6) If depletion of microglia does not alter the number of axons, but just their branching, why do manipulations of complement seem to alter total numbers of axons? The argument made in the Discussion that in this case there are compensations is possible, but other mechanisms should be considered, in particular the possibility that C3 has an impact on microglia-mediated neuron elimination via apoptosis (see paper about RGC apoptosis and C3r, Anderson et al., 2019).

7) The authors should present the criteria that define trogocytosis in the manner used in this manuscript, especially as there is no EM data to support its morphological assessment.

8) The images in Figure 7 do not appear to be representative of the data in the accompanying graph for branch numbers.

9) Several citations are referred to incorrectly: Weinhard et al., 2018 presented evidence for trogocytosis in fixed brain tissue using electron microscopy; Paolicelli et al., 2011 did not investigate complement factors.

Reviewer #3:

The manuscript "Microglial trogocytosis and the complement system regulate axonal prunning in vivo" by Lim and Ruthazer is a very elegantly designed and written paper showing for the first time in vivo evidence of microglial trogocytosis, using the *Xenopus* tadpole retinotectal circuit as a model.

The paper is very solid and I only have a few comments.

1) The authors seem to consider that microglial synaptic pruning is well-established whereas axonal trogocytosis is not but this is not exactly the case. The evidences for microglial pruning synapses come from indirect studies using KO and depletion models – these studies demonstrate a role for microglia but do not provide direct evidence that microglia actively selects the synapses to be removed (in opposition to neurons shedding off the synapses, later engulfed by microglia), although this is widely stated in the literature. Live imaging studies have failed to show that microglia grasp and digest synapses from living neurons. In fact, the Weinhard and Gross paper using organotypic cultures was unable to observe this phenomenon and instead reported axonal trogocytosis.

The authors should make a more critical appraisal of the literature regarding microglial synaptic pruning, as in fact it further highlights the relevance of their in vivo model for live imaging studies of microglial interaction with spines and axons.

2) This reviewer is a bit skeptic about the basal green autofluorescence in microglial cells and the ability of the authors to discriminate it from true "trogocytosis" signal. Would it be possible to perform some sort of quantification with appropriate negative controls (for instance, experiments with unlabeled axons, or with off-target labeling in other brain structures) to demonstrate the specificity of their quantification?

Additional evidence of the specificity of their analysis could be provided by doing correlational EM analysis of the microglia imaged by 2p.

Finally, the results will be more solid if the authors could quantify the length of the axonal arbor before and after microglial contact – to estimate the size of the arbor removed.

3) In the microglial depletion experiments the authors should more explicitly discuss that their manipulation is likely to affect microglia outside the optic tectum, including the retina, regions involved in locomotion, as well as peripheral macrophages. They should thoroughly discuss and provide as many evidences as possible that the depleted animals do not have impairments in their health status, vision, or locomotor activity (this is slightly touched upon in the Discussion), or any other factor that may affect their performance in the behavioral tests.

4) In all the aRCA3 overexpression and VAMP2-C3 expression experiments (Figures 7 and 8), the authors should show the direct evidence that trogocytosis is altered by quantifying the axonal fluorescent signal in microglia (as in Figure 3C), as proof that their manipulation had the predicted effect of potentiating or decreasing engulfment (instead of showing only the downstream effect in axons).

[Editors' note: further revisions were suggested prior to acceptance, as described below.]

Thank you for submitting your article "Microglial trogocytosis and the complement system regulate axonal pruning in vivo" for consideration by *eLife*. Your article has been reviewed by two peer reviewers, and the evaluation has been overseen by a Reviewing Editor and K VijayRaghavan as the Senior Editor. The following individuals involved in review of your submission have agreed to reveal their identity: Carlos Aizenman (Reviewer #1); Amanda Sierra (Reviewer #3).

The reviewers have discussed the reviews with one another and the Reviewing Editor has drafted this decision to help you prepare a revised submission.

Summary:

Your study on microglial pruning of axon endings through partial engulfment is excellent and well documented. Using the developing retinotectal projection of *Xenopus laevis* tadpoles as a model system, you have presented a varied set of experiments that include molecular manipulations, live imaging and behavior, to provide compelling in vivo evidence that microglia prune axons via trogocytosis – a partial engulfment of an axon and breakdown of axon terminals, identifying this process as an important regulator of axonal pruning to promote correct neural wiring during development.

Especially pointed is the experiment to overexpress aRCA2 (equivalent to human CD46) and thus inhibiting complement, when you see decreased microglial trogocytosis and enhanced axonal branching, whereas increasing complement expression in axons (by anchoring it to the presynaptic terminal) you see increased pruning. Together with the functional experiments, these data show compelling evidence that the complement system regulates activation of microglia, and thus axonal pruning, which results in normal sculpting of axonal projections in the developing brain.

Your revisions are apt and thorough, and address the concerns of the reviewers. Reviewer 1 in fact said, "(this is) probably one of the best work I've seen from this group, already known for excellent, high-quality papers in developmental neuroscience."

Revisions:

Reviewer 3 had two caveats and you can address these by textual revision:

1) Refrain from using the term "activated microglia" or "microglial activation" since there is no unique state of activation. Rather, microglia show different responses to different stimuli. As you show, in physiological conditions, they are far from "resting" and play critical roles by responding to neuronal cues.

2) This reviewer had initial concern about the specificity of your novel imaging method, which you largely addressed with new figures. However, you have not provided additional evidence of engulfment with an independent method, such as correlational EM analysis, as this reviewer recommended. You lightly touched on this issue by mentioning the EM study performed by Weinhard et al. , but a more explicit acknowledgement of the need of independent corroboration of their findings is still needed.

---

## [Author Response]

Revisions:1) Measuring changes in branching versus axon length: Reviewer 2 questioned whether microglia depletion results only in a change of branching, but not overall length of axons. During the reviewer discussion, reviewer 1 pointed out that the changes observed occur over a short time course, of small lengths of an otherwise large arbor and so changes in axon length would not be a useful measure. I agree. This point should be taken up in the Discussion.

The following text has been added to the Discussion:

“Another interesting possibility is that microglia preferentially engage in trogocytosis at short, dynamic, branches over longer, established, mature branches. In this case, microglial depletion results in a greater number of short branches, which over the short time course in this study, do not significantly impact axon branch length.”

2) Controls for autofluorescence, to demonstrate true trogocytosis signal vs autofluorescence:a) As suggested by reviewer 3, is it possible to quantify with appropriate negative controls (for instance, experiments with unlabeled axons, or with off-target labeling in other brain structures) to demonstrate the specificity of the quantification?b) Reviewer 2 suggests that traces for all individual cells in the validation experiments (Figure 2F) should be shown to understand the variability in the persistent change in GFP signal that remains after contacts are completed. In addition, a spectral analysis should be done to help rule out autofluorescence.c) Correlative EM, also suggested, would be a labor-intensive addition, and is not deemed necessary if one of the former suggestions are taken.

The important negative control experiment to address the question of autofluorescence, in which the experiment is performed with unlabeled axons, was actually integrated into our original experimental design in Figure 3C and D (i.e., “no axons” condition). No significant increase in green fluorescence was detected in this condition.

We have also added a supplemental figure (new Figure 2—figure supplement 1) which follows over time the changes in intensity of green fluorescence associated with each microglial cell that contacts (or fails to contact) a GFP-expressing axon. The stable levels of green fluorescence in non-interacting microglia lends further support to our finding that green fluorescence in microglia increases specifically as a result of axon-microglia contact.

In addition, we have added a new quantitative analysis (new Figure 3—figure supplement 1) which examines green fluorescence across all individual cells in the trogocytosis assay validation experiment.

3) Complement experimentsa) Reviewer 2 thought that the proper control for the VAMP2-C3 overexpression is VAMP2 itself, is not ideal, citing that VAMP2-C3 fusion might have dominant negative effects that could have little to do with C3; also, that a better control would be VAMP2-C3* where C3* is non-cleavable or even a VAMP2-GFP fusion. He also wondered why you did not use a GPI anchor if the idea was to present C3 on the membrane, and whether you had evidence that C3 in this fusion form is functional. In the Consultation session, reviewer 1 did not think that the VAMP-GFP fusion or uncleaveable C3 would be better, and surmised that you used VAMP rather than GPI anchors to limit expression to presynaptic terminals. Reviewer 2 accepts this point but please discuss the inherent weaknesses in using the VAMP2-C3 control, either in the Materials and methods and/or Results.

We added to the Results to explain why we didn’t use a GPI anchor design:

"This design was chosen in favor of a GPI anchor design that modifies the C-terminus C345C domain of complement C3 as this domain undergoes major rearrangement during activation and proteolysis (32° rotation, 10 Å translation) (Janssen et al., 2005). In contrast, the N-terminus of complement C3 is exposed on the surface of the protein and located on the MG1 domain, a domain that does not undergo marked confirmational changes upon complement C3 activation and proteolysis (3° rotation, 1 Å translation).”

Also added to the Discussion to discuss the shortcomings of the VAMP2 control:

“While VAMP2-C3 clearly produced effects on axon morphology, we did not rule out the possibility that the expression of VAMP2-C3 in RGC neurons may be producing effects on axons that are not mediated by the complement system.”

b) Reviewer 3 thought that for the aRCA3 overexpression and VAMP2-C3 expression experiments (Figures 7 and 8), you should show direct evidence that trogocytosis is altered by quantifying the axonal fluorescent signal in microglia (as in Figure 3C), as proof that engulfment is potentiated or decreased (instead of showing only the downstream effect in axons). Reviewer 1 rebutted this comment by indicating that he did see a difference in axon phenotypes for depletion and CD46 homolog or C3 overexpression. He also acknowledges that you do discuss and invoke compensations due to the cell autonomous nature of the CD46 homolog and C3 over-expression. However, he did find it disappointing that you did not follow up on a single axon level with longitudinal in vivo live imaging what the impact of microglia contacts is in these circumstances. Please address the shortcomings of these experiments in the Discussion.

Added to the Discussion:

"Additionally, we were unable to determine whether VAMP2-C3 expression enhanced microglial trogocytosis as electroporation of RGC neurons with VAMP2-C3 rarely yielded labeled axons. We speculate that expression of VAMP2-C3 in RGC neurons reduces viability, as previous studies have shown that the complement cascade mediates phagocytosis of RGC neurons during development (Anderson et al., 2019). Furthermore, our study indirectly examined the effects of VAMP2-C3 and aRCA3 on microglial trogocytosis. Future experiments could directly examine the effects of VAMP2-C3 expression and aRCA3 overexpression on trogocytosis rates by real-time imaging.”

c) Reviewer 2, # 6 comment on why depletion of microglia does not alter the number of axons, only their branching, but manipulations of complement seem to alter total numbers of axons. Please cite the possibility that C3 has an impact on microglia-mediated neuron elimination via apoptosis (see on RGC apoptosis and C3r, Anderson et al., 2019). Reviewer 1 thought that both microglia depletion and complement manipulation altered number of axonal branches, not number of axons. Please address this point in the Discussion.

We changed the Discussion to be clearer that the number of axons was not examined in this study. Our experiments select for labeling of isolated pHtdGFP-expressing single axons that can be reconstructed for morphometric analysis. This experimental design therefore does not provide direct or indirect information about the total number of retinal axons.

4) RGC subtype specificity; pruning in areas outside the tectum:a) As per reviewer 1's comments, that because there is a bigger effect on dark looming detectors than light looming detectors, different types of axons (from different RGCs having different response selectivity) could be differently selective to microglial pruning.b) Inhibition of microglial activation by PLX5622 might also affect remodeling in visual circuits, and could account for the behavioral difference.Please discuss.

Added to the Discussion:

“Conversely, it is possible that dark and bright looming-sensitive RGC axons are differentially affected by microglial depletion or the circuitry of the retina upstream of RGCs is remodeled following microglial depletion, both of which could produce the abnormal behavioral outcomes we observed.”

5) Unease with the behavioral results: Reviewer 2, comment #4 and Figure 9: the "behavioral part of the manuscript is only weakly linked to the main story". Reviewer 1 somewhat concurs, especially since this behavior is something that is known to require intact tectal circuitry. To this end, reviewer 3 proposes that to ensure that microglia outside the optic tectum, including the retina, regions involved in locomotion, as well as peripheral macrophages are not affected, evidence should be provided that the depleted animals do not have impairments in their vision, locomotor activity, and overall health. Reviewer 2 subsequently agreed to include the behavior, provided that you explicitly state that the mechanism that links depletion to behavior is far from understood, and remove Figure 9 summary, as it now stands, "to offset the abundance of correlative findings and hype in this field and reinforcement of (such) data". Reviewer 3 adds that you could modify Figure 9 to summarize what is known and what is not, and put question marks whenever appropriate. The latter would be a compromise position.

We concur and have removed Figure 9C which relied too heavily on correlative findings and added to the Discussion:

"Indeed, the mechanism linking microglial depletion to improper response to looming stimuli is unclear and warrants future investigation.”

6) Literature citation: All three reviewers ask for a more critical appraisal of the literature regarding microglial synaptic pruning, to highlight what you have done so well and how it is novel (see Reviewer 2, point #9; Reviewer 3: Point #1): Although previous and widely cited studies argue for microglial pruning, based on KO and depletion models, there is little direct evidence that microglia actively select the synapses to be removed (in contrast to neural shdding of synapses, later to be engulfed by microglia). Other live imaging studies have failed to show that microglia grasp and digest synapses from living neurons; this includes the Weinhard and Gross study using organotypic cultures. They instead report axonal trogocytosis, for the first time, in vivo. Yours is the first study very nicely demonstrating trogocytosis by live imaging.

We added to the Introduction:

“Studies depleting microglia or disrupting microglial function provide indirect evidence to support the hypothesis that microglia remodel synapses through phagocytic mechanisms. An inherent weakness of indirect approaches is that the source of the synaptic material within microglia is unknown. For example, it is possible that synaptic components may be found within microglia due to clearance of apoptotic neurons rather than synaptic pruning. More direct approaches are required to verify whether microglia collect synaptic material from living neurons.”

Modified the main text to be more assertive about trogocytosis results.

Modified the Introduction to include the mention of electron micrographic data in the Weinhard et al., 2018 study.

Removed incorrect references attributing complement studies to Paolicelli et al.

[Editors' note: further revisions were suggested prior to acceptance, as described below.]

Revisions:Reviewer 3 had two caveats and you can address these by textual revision:1) Refrain from using the term "activated microglia" or "microglial activation" since there is no unique state of activation. Rather, microglia show different responses to different stimuli. As you show, in physiological conditions, they are far from "resting" and play critical roles by responding to neuronal cues.

We have removed both of these references to microglial activation from the manuscript.

2) This reviewer had initial concern about the specificity of your novel imaging method, which you largely addressed with new figures. However, you have not provided additional evidence of engulfment with an independent method, such as correlational EM analysis, as this reviewer recommended. You lightly touched on this issue by mentioning the EM study performed by Weinhard et al., but a more explicit acknowledgement of the need of independent corroboration of their findings is still needed.

The first paragraph of the Discussion section, summarizing our results now explicitly acknowledges that our finding was not confirmed using correlative EM:

“As SYP-pHtdGFP is primarily localized to synaptic vesicles, this suggests that the axonal material that microglia engulf contains presynaptic vesicles and is consistent with correlative electron microscopy studies that have demonstrated putative presynaptic vesicles within microglia (Weinhard et al., 2018), although we did not use correlative electron microscopy to confirm this in our study.”